*Method*

# The GENDULF algorithm: mining transcriptomics to uncover modifier genes for monogenic diseases

Noam Auslander[1,2,†,§] iD, Daniel M Ramos[3,†], Ivette Zelaya[4], Hiren Karathia[5,§] iD,

Thomas O. Crawford[6,7], Alejandro A Schäffer[1,§] iD, Charlotte J Sumner[3,7,‡] iD & Eytan Ruppin[1,*,‡,§] iD

## Abstract

**Modifier genes are believed to account for the clinical variability observed in many Mendelian disorders, but their identification remains challenging due to the limited availability of genomics data from large patient cohorts. Here, we present GENDULF (GENetic moDULators identiFication), one of the first methods to facilitate prediction of disease modifiers using healthy and diseased tissue gene expression data. GENDULF is designed for monogenic diseases in which the mechanism is loss of function leading to reduced expression of the mutated gene. When applied to cystic fibrosis, GENDULF successfully identifies multiple, previously established disease modifiers, including *EHF*, *SLC6A14*, and *CLCA1*. It is then utilized in spinal muscular atrophy (SMA) and predicts *U2AF1* as a modifier whose low expression correlates with higher SMN2 pre-mRNA exon 7 retention. Indeed, knockdown of U2AF1 in SMA patient-derived cells leads to increased full-length SMN2 transcript and SMN protein expression. Taking advantage of the increasing availability of transcriptomic data, GENDULF is a novel addition to existing strategies for prediction of genetic disease modifiers, providing insights into disease pathogenesis and uncovering novel therapeutic targets.**

**Keywords** cystic fibrosis; digenic inheritance; gene expression; modifier gene; spinal muscular atrophy

**Subject Categories** Computational Biology; Genetics, Gene Therapy & Genetic Disease; Molecular Biology of Disease

**Mol Syst Biol. (2020) 16: e9701**

## Introduction

Phenotypic heterogeneity is observed in many Mendelian diseases such that patients with the same mutation may develop a severe form of disease, a mild one, or show no symptoms at all. Among the factors that account for these differences are *modifier genes* (Nadeau, 2001), whose activity influences disease severity. Identifying such genes has major implications for disease prognostication and development of novel therapeutics (Antonarakis & Beckmann, 2006). However, due to the low frequency of the mutations causing most Mendelian disorders and the scarcity of large relevant patient cohorts, only a few modifier genes have been identified thus far (Génin *et al*, 2008; Kousi & Katsanis, 2015) leaving the mechanisms underlying clinical variability of most Mendelian disorders poorly understood.

Existing strategies for studying the role of genetic factors in determining phenotypic presentation are often classified into three categories depending on the type of data analyzed (Génin *et al*, 2008; Kousi & Katsanis, 2015): (i) genome-wide association studies (GWAS), which compare the distribution of marker genotypes in patients with different disease phenotypes, (ii) genetic linkage studies, using tools such as Superlink (Fishelson & Geiger, 2002) and GENHUNTER-TWOLOCUS (Dietter *et al*, 2004), which assess the inheritance pattern and content of alleles shared between phenotypically concordant and discordant relatives, and (iii) systematic genome-wide exome sequencing projects, which identify individuals who are resilient to otherwise phenotype-causing mutations. A recent example of the third approach is the Resilience Project that analyzes genomes to ascertain subjects who are healthy despite harboring disease-causing mutations (Chen *et al*, 2016). Each of these existing strategies as well as disease-specific wet laboratory functional screens may yield a long list of candidate modifier genes, emphasizing the need for complementary methods to narrow in on a smaller set of candidate modifiers to validate.

1 Cancer Data Science Laboratory (CDSL), National Cancer Institute, National Institutes of Health, Bethesda, MD, USA
2 National Center for Biotechnology Information, National Library of Medicine, National Institutes of Health, Bethesda, MD, USA
3 Department of Neuroscience, Johns Hopkins University School of Medicine, Baltimore, MD, USA
4 Interdepartmental Program in Bioinformatics, University of California Los Angeles, Los Angeles, CA, USA
5 Laboratory of Receptor Biology and Gene Expression, National Cancer Institute, National Institutes of Health, MD, USA
6 Department of Pediatrics, Johns Hopkins University School of Medicine, Baltimore, MD, USA
7 Department of Neurology, Johns Hopkins University School of Medicine, Baltimore, MD, USA
 *Corresponding author. Tel: +1 240 858 3169; E-mail: eyruppin@gmail.com
 †These authors contributed equally to this work as first and second authors
 ‡These authors contributed equally to this work as last authors
 §This article has been contributed to by US Government employees and their work is in the public domain in the USA.

Here, we present a new approach for the genome-wide identification of genetic modifiers of monogenic disorders, termed GENetic moDULators identiFication (GENDULF). We use the term "monogenic disease" for disorders in which mutations in one gene determine who is affected with high penetrance, but variations in that gene alone may not fully explain the variable phenotypes seen in different patients. The GENDULF method is intended for monogenic diseases in which the mechanism is loss-of-function or reduced function. For many of these diseases with known gene modifiers (Gazzo *et al*, 2016), mutations of the gene causal of disease (herein termed GCD) often result in its reduced expression. We therefore reasoned that some healthy individuals who have reduced expression of a GCD are protected from disease by differential expression of other genes (the modifiers). We do not assume that every disease-causing mutation in the GCD leads to reduced gene expression, but rather that some of them do. Under this assumption, reduced expression of the GCD can be deleterious, even if it does not occur in the context of a disease-causing mutation. The interactions we seek between the GCD and modifiers are akin to some definitions of the genetic term "epistasis", but we avoid using this term because it is sometimes associated with formal measures of overall organism fitness, which we do not compute.

GENDULF operates by mining gene expression data of healthy tissues, available most prominently from the Genotype Tissue Expression project (GTEx; https://gtexportal.org), and disease-vs.-control tissues to identify expression patterns of genes that may modify disease severity. The GENDULF approach is feasible because gene expression in healthy individuals can vary significantly due to genetic and non-genetic reasons (Curtis *et al*, 2012). Variation in the expression of a GCD may be explained, at least in part, by regulation of other genes that compensate for low expression levels of the GCD in healthy individuals particularly in the tissues most relevant to the disease etiology. Because GTEx contains tissue-specific gene expression and for some diseases, the most affected tissues are known, GENDULF can examine gene expression in the disease-relevant tissues of healthy subjects. By identifying unaffected GTEx individuals with very low tissue-specific expression levels of the GCD, GENDULF can predict potential modifiers that may compensate for these low GCD levels. The expression levels of these potential modifier genes are then examined in disease-relevant tissues from affected and unaffected individuals to evaluate the association of candidate modifiers with the disease phenotype. Since we are interested in identifying 'actionable' modifiers, which could most readily be targeted by drugs to inhibit their activity, we focus on *negative modifiers*—genes whose inactivation could alleviate the disease phenotype. Nevertheless, GENDULF could be readily modified to identify targets whose increased expression may alleviate disease phenotypes.

We first tested the ability of GENDULF to identify tissue-specific modifiers of cystic fibrosis (CF) because it is the most common recessive Mendelian disease and has variable severity. CF is caused by biallelic mutations of the cystic fibrosis transmembrane conductance regulator gene [*CFTR* (Rommens *et al*, 1989)], resulting in disrupted epithelial fluid transport in lungs, pancreas, colon, and other organs (Cutting, 2015). A twin study suggested that 50% of the variability in CF lung function is due to genetic factors (Collaco *et al*, 2010). Several CF modifiers have been previously discovered in both lung and colon tissues (Wright *et al*, 2011; Gallati, 2014; Corvol *et al*,

2015) providing an opportunity to evaluate the GENDULF approach for a disease in which there are established results.

To determine whether GENDULF can be utilized in other diseases, we then applied it to spinal muscular atrophy (SMA), a neuromuscular disorder of variable severity caused by biallelic mutations of the survival motor neuron 1 gene (*SMN1*) (Lefebvre *et al*, 1995) and retention of the paralog gene *SMN2*. A cytosine to thymine nucleotide change in exon 7 of *SMN2* leads to frequent exclusion of exon 7 during splicing of SMN2 pre-mRNAs and thus less functional SMN protein (Monani *et al*, 1999). We applied GENDULF to SMA to search for candidates that may influence *SMN2* exon 7 pre-mRNA splicing, as this is an important determinant of disease severity (Prior *et al*, 2009; Hua *et al*, 2011; Wu *et al*, 2017).

Together, our findings support the utility of GENDULF in prioritizing disease modifiers in CF and SMA. Particularly, when used in conjunction with available GWAS or relevant biological insights, and when transcriptomic data are available, GENDULF could facilitate identification of modifier genes for other loss-of-function Mendelian diseases.

# Results

### The GENDULF approach

We provide an overview of GENDULF here and refer the reader to a full description in Materials and Methods below. GENDULF consists of two steps at its core (Fig 1, Materials and Methods). [Step 1] The aim of the first step is to find genes that are downregulated when the GCD is downregulated in *healthy individuals* and particularly in *the tissues that are relevant to the disease* in question. We reason that in healthy individuals with very low tissue-specific expression of the GCD, compensatory downregulation of some other genes may in part help maintain the observed unaffected phenotype. We term the candidate genes *Potential Modifiers* (*PMs*). [Step 2] The aim of the second step is to find genes that are not downregulated when the GCD is downregulated in disease-relevant tissues of individuals *affected with the disease*, thus testifying that their co-expression with the GCD is specific in healthy individuals and may have compensating effect. In step 2, we examine expression levels of the PMs in data sets that include both disease and control samples. We define *disease-associated PMs (DPMs)* as those PM genes that lose the association with the GCD (found in the tissue of unaffected individuals) in the relevant tissues of affected individuals; that is, they are not significantly downregulated in disease samples in which the GCD is, by definition, inactive. The downregulation of a PM in the healthy controls and the absence of downregulation of the same PM in patients rules out the possibilities that either the PM is generally co-regulated with the GCD or downregulated in a signaling pathway downstream of the GCD. As demonstrated in the analysis of SMA, when the disease phenotype is influenced by expression of a known specific modifying transcript, a third step may be introduced, as described later in that case.

### Applying GENDULF to identify gene modifiers of CF

To evaluate the performance of GENDULF, we first applied it to a monogenic disorder in which several genetic modifiers have been

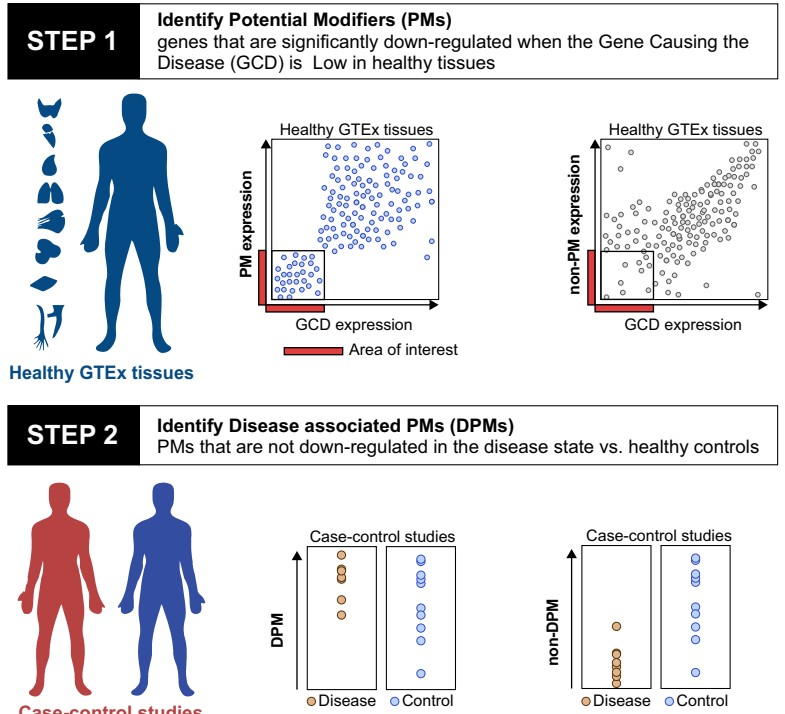

**Figure 1.  An overview of GENDULF computational approach.**

The two steps of GENDULF: (Step 1) Mine transcriptomics of healthy disease-relevant tissues to identify PMs, which are genes that are differentially under-expressed when the GCD is lowly expressed. GENDULF does not compute a correlation across the whole range of expression values, but specifically searches only for a significant association at the lower level range, as shown in the region boxed in the left scatter plot, where the dots are in blue and the *y*-axis is labeled 'PM' for 'potential modifier'. The scatter plot on the upper right depicts an example of relationship between expression of the GCD and another gene in which GENDULF is not expected to find the other gene as a modifier, and hence the y-axis is labeled non-PM expression. (Step 2) Evaluate the expression of PMs identified in step 1 in transcriptomic data sets containing both diseased and control samples to find a subset of PMs that we label as DPMs (left graphs)—PM genes that are not down regulated in the disease tissues.

previously identified (Drumm *et al*, 2005; Cutting, 2010; Wright *et al*, 2011). CF is caused by biallelic, loss-of-function mutations of *CFTR* and affects 60,000 individuals worldwide (Kerem *et al*, 1990). It is clinically characterized by mucous retention in the lungs, pancreas, colon, and other organs and repeated lung infections result in significant morbidity and mortality (O'Sullivan & Freedman, 2009). It is a good model for the identification of disease modifiers because it has a relatively high prevalence leading to availability of data on many accessible patients for performing detailed phenotypic analyses. Furthermore, while a large fraction of patients are heterozygous or homozygous for the ΔF508 mutation, these patients can exhibit quite divergent phenotypes (Drumm *et al*, 2005), where several other mutations cause disease via lowering of CFTR expression; consistent with the premise of GENDULF (Kerem *et al*, 1997; Ramalho *et al*, 2002). In our first test of GENDULF, we focused on CF patients with either lung or intestinal disease, to mitigate the effects of allelic heterogeneity, as was done in other studies seeking CF modifiers (Drumm *et al*, 2005; Stanke *et al*, 2010).

To identify modifiers of lung disease in CF, we employed Step 1 of GENDULF to RNA sequencing data derived from 320 healthy lung samples deposited in the GTEx database, which includes transcriptomics data from 53 tissues and 544 human subjects (The GTEx Consortium, 2013). This step pointed to 366 PM genes that were found significantly downregulated when CFTR was

downregulated in healthy lung tissues (Dataset EV1). Applying Step 2 of GENDULF, we examined the expression of each of these PM genes in nasal brushings of the inferior turbinates of mild and severe CF patients (identical homozygote ΔF508) and healthy controls (Wright *et al*, 2006). The expression of most PMs was decreased in CF patient tissue compared with healthy tissue in a similar pattern to CFTR, indicating that their expression may simply be correlated with that of CFTR activity. However, 131 (36%) passed GENDULF step 2 and were found not to be similarly downregulated when CFTR expression was impaired in CF tissues (Materials and Methods, Dataset EV1) testifying to their potential compensatory role as DPMs. Examining all reported CF genetic modifiers of the lung phenotype we collected from the literature (Dataset EV2), we find that the GENDULF-predicted DPMs are highly enriched with previously verified modifiers of CF manifestations in the lung (with eight overlapping genes, *P*-value = 5.4109e-15 from a hypergeometric test (Johnson & Kotz, 1977), Dataset EV2).

We found that 8 of the 10 previously published CF lung modifiers that passed GENDULF step 1 also passed GENDULF step 2; the two exceptions (*SFTPA1*, *SLC26A9*) had no expression measurements in the case–control data and hence could not possibly be detected at step 2. All four previously published colon CF modifiers that passed GENDULF step 1 were eligible at step 2 and also passed step 2. We

conclude that GENDULF is effective at finding those known PMs that fit the expected pattern of expression in GTEx (Fig 1, upper) and are measured in the case–control gene expression data. A recent review of CF lung phenotype modifiers found evidence in favor of two additional predicted DPM genes (Dataset EV2), namely *KRT8* and *MUC1* (Shanthikumar *et al*, 2019). Some of these known modifiers are ranked within the top DPMs, including *EHF*, *SFTPA2*, and *SLC6A14*, all previously identified modifiers of lung disease severity in CF (Choi *et al*, 2006; Tagaram *et al*, 2007; Wright *et al*, 2011; Li *et al*, 2014) (Fig 2A–C). The number of samples in this dataset with mild lung manifestations is too small to evaluate whether modifiers are differentially expressed between CF patients with mild lung disease and severe lung disease. Importantly, we also find that these four CF predicted modifiers (*CLCA1*, *FABP1*, *MUC2*, *SLC4A4*) tend to be co-downregulated in those GTEx lung samples in which CFTR expression is low (all pairwise hypergeometric *P*-values evaluating the overlap between pairs of these modifiers are < 0.05, Fig 2D).

To estimate the robustness of these results, we apply a sensitivity, specificity, and positive predictive value (PPV) analysis of the GENDULF-predicted modifiers against these previously verified modifiers, when setting different thresholds for GENDULF step 1 (See Materials and Methods for details, Appendix Fig S1). The specificity is very high and close to the perfect 1.0, but this is partially driven by the small number of known modifiers (positives). The sensitivity is not very high, but still over two orders of magnitude higher than would be expected by chance, providing a manageable rate of modifiers that are predicted with GENDULF. The PPV is also substantially higher than would be expected by chance.

To estimate the number of case–control samples that are required for GENDULF step 2 given GENDULF step 1 results, we provide a power calculation capability in the software (see Materials and Methods). We evaluated the power for the GENDULF step 1 modifiers obtained for CF lung disease and find, for example, that seven cases and seven controls are expected to be sufficient to have 80% power to detect a modifier (See Materials and Methods for details and Appendix Fig S2).

To identify modifiers of intestinal disease in CF, we applied GENDULF to analyze 345 non-CF colon samples from the GTEx database and examined the expression of these genes in rectal mucosal epithelia from CF patients (bearing ΔF508 mutation) and healthy controls (Stanke *et al*, 2014). From 344 PM candidates identified in Step 1 in the healthy colon tissue, 123 (35%) are predicted to be DPMs in step 2, i.e., their expression is not significantly downregulated when CFTR was mutated in tissues from CF patients (Dataset EV1). Examining reported CF genetic modifiers of colon disease collected from the literature (Dataset EV2), the top GENDULF-predicted modifiers are highly enriched with those previously reported (with four overlapping genes, hypergeometric test *P*-value = 3.9311e-08, Dataset EV2). These include *CLCA1*, whose locus has been reported to modulate gastrointestinal defect in CF (Ritzka *et al*, 2004; Van Der Doef *et al*, 2010) (Fig 2E) and *SLC4A4*, which was found to modify the intestinal phenotype in CF (Dorfman *et al*, 2009) (Fig 2F). Furthermore, the genes identified by GENDULF are highly enriched with genes located on chromosome 19q13 (hypergeometric enrichment *P*-value = 9e-04, Dataset EV2), a locus associated with variability in the CF colon phenotype (meconium ileus) (Zielenski *et al*, 1999). Some studies have suggested that the linkage signal in 19q13 is best explained by the functional candidate gene *TGFB1*

(Bremer *et al*, 2008; Corvol *et al*, 2008), but a fine mapping study provided evidence in favor of a tight cluster of three other immune-related genes in 19q13: *CEACAM5*, *CEACAM3*, *CEACAM6* (Stanke *et al*, 2010). One of the candidates found by GENDULF in this band is *CEACAM5*, supporting the latter hypothesis.

## Combining GENDULF with association or linkage studies

A particular strength of GENDULF is its ability to be used in conjunction with prior genomic data to help guide discovery of modifier genes. We studied the incorporation of GENDULF with data from association studies and its application to a given list of genes in a locus. In one study, five independent genomic loci were shown to have significant associations with variation in the clinical severity of CF lung disease (Corvol *et al*, 2015). Yet, as such association loci typically include numerous genes, the identification of the individual genes within the loci that correlate to disease variation can be challenging (Génin *et al*, 2008). We applied GENDULF to evaluate each of the genes contained within these five loci. To this end, for each gene in a given locus, we evaluated the level by which its expression is significantly downregulated when the expression of CFTR is low in non-disease lung tissues (GENDULF step 1). We found that at four of these five loci (all except chr5p15.3) at least one gene is significantly downregulated when the expression of CFTR is low (Fig 2G, Appendix Fig S3A). Notably, the genes having the maximal co-downregulation levels with CFTR coincide with the strongest modifiers reported in the literature (Wright *et al*, 2011; Corvol *et al*, 2015) including *EHF*, *MUC4*, *HLA-DRA*, and *SLC6A14*. A robustness analysis shows that the number of genes pinpointed by GENDULF that are also previously published modifiers is more than would be expected by random chance, as it is never obtained when randomly shuffling the CFTR expression values (Materials and Methods). Evaluating GENDULF Step 1 for three loci found associated with CF intestinal phenotype (Sun *et al*, 2012), we find a similar trend to the one observed for lung loci (Fig 2G), but considerably less significant (Appendix Fig S3B). Taken together, these results show that GENDULF can be applied successfully in a hybrid manner in which GWAS or linkage analysis is used to first identify a genomic locus containing numerous candidate modifier genes, and GENDULF is then applied to those candidate genes to find genes whose low expression is most associated with low GCD expression in healthy tissues.

## Applying GENDULF to identify gene modifiers of SMA

We next applied GENDULF to predict novel genetic modifiers of SMA, a neuromuscular disease that is the leading inherited cause of infant mortality and is characterized by degeneration of α motor neurons in the spinal cord as well as atrophy and weakness of skeletal muscles (Crawford & Pardo, 1996). SMA can manifest with variable severity, but is always caused by recessive mutations of the survival motor neuron 1 (*SMN1*) gene (Brzustowicz *et al*, 1990; Lefebvre *et al*, 1995), which is the GCD. *SMN1* mutations are often genomic deletions leading to loss of SMN1 mRNA expression. Embryonic lethality is prevented in SMA patients by retention of the *SMN2* paralog gene in variable copy number (Burghes & Beattie, 2009). SMN2 pre-mRNAs undergo alternative splicing, often excluding exon 7, resulting in a transcript that encodes a truncated SMN

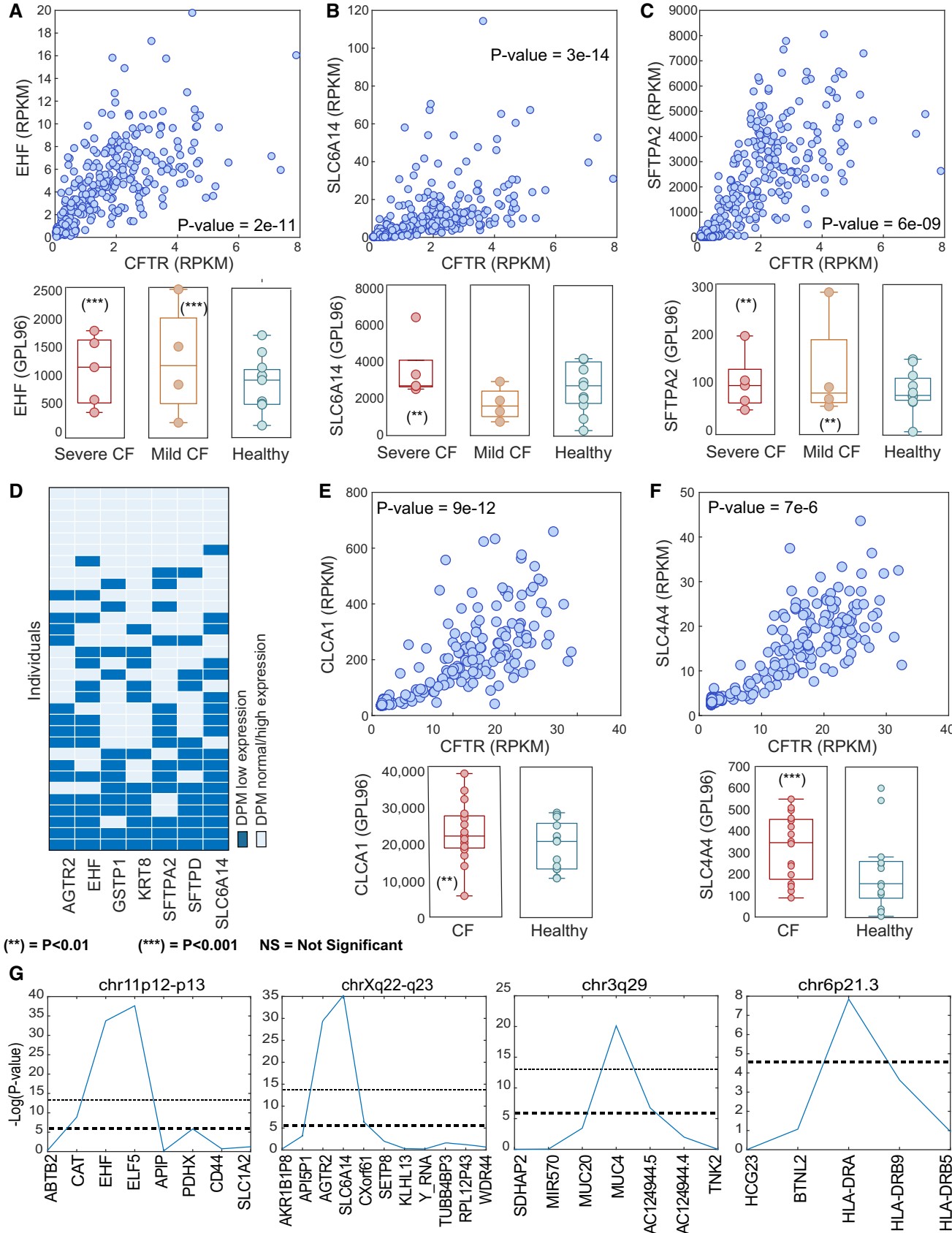

Figure 2.

**Figure 2.   CF modifiers identified by GENDULF.**

A–C   Upper panels: Scatter plots associating the expression of the GCD (CFTR) vs. identified PMs (A) EHF, (B) *SLC6A14*, and (C) *SFTPA2* in healthy lung tissues from GTEx. Bottom panels: Boxplots associating the expression of the identified DPMs in case–control studies; the expression of (A) EHF, (B) *SLC6A14*, and (C) *SFTPA2* in severe and mild CF and in healthy controls. For all boxplots, center lines indicate medians, box edges represent the interquartile range, whiskers extend to the most extreme data points not considered outliers, and the outliers are plotted individually. Points are defined as outliers if they are greater than $q_3 + w \times (q_3 - q_1)$ or $< q_1 - w \times (q_3 - q_1)$, where $w$ is the maximum whisker length, and $q_1$ and $q_3$ are the 25th and 75th percentiles of the sample data, respectively. There are five severe (red), four mild (orange), and eleven control (blue) biological replicates.

D   Map of coincidence of the low expression (lowest 10% of expression levels) of the 7 top DPMs with the low expression of CFTR in GTEx lung samples (lowest 10% of expression levels, when symptoms would be present in CF patients Ramalho *et al*, 2002; Kerem *et al*, 1997); each small rectangle inside the big rectangle represents one individual; all presented samples are those with low CFTR expression. Dark blue rectangles indicate samples with low expression of the listed DPM).

E, F   Upper panels: scatter plot associating the expression of the GCD (CFTR) vs. identified PMs in healthy colon GTEx tissue; the expression of CFTR (*x*-axis) vs. that of (E) *CLCA1* and (F) *SLC4A4*, respectively, in healthy colon tissues. Bottom panels: Boxplots associating the expression of the identified DPMs in case–control studies; the expression of (E) *CLCA1* and (F) *SLC4A4* in colon tissues from CF and healthy controls. Empirical *P*-value significance is indicated qualitatively for two thresholds. There are sixteen CF (red) and thirteen control (blue) biological replicates.

G   The *P*-values assigned by GENDULF to genes within chr11p12-p13, chr6p21.3, chr3q29, and chrXq22-q23 chromosomal segments, ordered by their location. The lower dashed line represents a significance threshold corrected for the number of genes evaluated, and the upper dashed line represents a significance threshold corrected for all genes and transcripts in GTEx, with alpha = 0.05.

Data information: **P-value < 0.01 and ***P-value < 0.001, using the permutation test defined in the Materials and Methods section. The *P*-values in panels (A, B, C, E, F, and G) are for the hypergeometric enrichment test.

protein that is rapidly degraded (Burnett *et al*, 2009; Cho & Dreyfuss, 2010). Enhanced inclusion of exon 7 into mature SMN2 mRNAs is associated with increased SMN protein levels and reduced disease severity (Prior *et al*, 2009; Hua *et al*, 2011; Finkel *et al*, 2016). Hence, SMN2 constitutes an attractive target for SMA therapy, where increased expression of full-length SMN2 is desired. We first applied GENDULF Step 1 to GTEx transcriptomics data derived from healthy muscles ($n = 430$) and subsequently to spinal cords ($n = 79$). While these are healthy tissues, we observe variations in SMN1 and SMN2 expression, which could be due to underlining variations in *SMN1* and *SMN2* copy number. We find 484 PMs that are significantly low when SMN1 mRNA expression levels are low in these tissues (lowest 10% of expression levels, Dataset EV3). We next applied GENDULF Step 2 to muscle and spinal cord tissues derived from SMA patients vs. healthy controls, which yielded 296 DPMs (Dataset EV3). Among these DPMs are several previously reported *negative* modifiers of SMN2 splicing (for which downregulation is reported to enhance exon 7 retention), such as *HNRNPU*, *SF1*, and *SRSF4* (Xiao *et al*, 2012; Wee *et al*, 2014) (Fig 3A–C).

SMN2 exon 7 inclusion serves as a unique modifier of SMA. Therefore, we added a transcript-specific third step to GENDULF (Materials and Methods). First, we investigated whether exon 7 inclusion may also compensate for low levels of SMN1 in healthy individuals. Indeed, we find that the ratio between full-length SMN2 (SMN2-FL, i.e. containing exon 7) to truncated SMN2 (SMN2Δ7, i.e. lacking exon 7) is increased in *healthy individuals* when SMN1 expression is low (Appendix Fig S4A and B). Given this pattern, we hypothesized that enhanced exon 7 inclusion and thus increased SMN2-FL expression may act as a compensatory rescue mechanism in healthy tissues when SMN1 expression is very low. Supporting this notion, we find that several previously reported modifiers that specifically enhance SMN2 exon 7 inclusion show significantly higher expression in healthy tissues with a high SMN2-FL to SMN2Δ7 ratio compared to tissues with a low SMN2-FL to SMN2Δ7 ratio (Appendix Fig S4C). Thus, in Step 3 of the GENDULF analysis we set out to identify, among the candidates identified from GENDULF steps 1–2, those that are most significantly associated

with SMN2 exon 7 inclusion, pinpointing *U2AF1* and *HNRNPA0* (Fig 3D and E). We then further evaluated these candidates experimentally. As controls (or potential *positive modifiers*, whose inactivation could aggravate the disease phenotype), we also evaluated *SF3B3* and *NECAP1*, as these were PMs identified in GENDULF step 1 but were not confirmed as DPM in steps 2 and 3 (Appendix Fig S5). To summarize, based on the GENDULF analysis, we predict that inactivation of U2AF1 and HNRNPA0 would enhance SMN2 exon 7 inclusion, whereas the inactivation of SF3B3 and NECAP1 would either reduce or have no effect over SMN2 exon 7 inclusion.

**Experimental validation of predicted SMA DPMs**

To investigate further the predictions of GENDULF, we knocked down the expression of the novel negative modifiers (*U2AF1* and *HNRNPA0*) and the neutral or positive modifiers (*SF3B3* and *NECAP1*, whose inactivation may be deleterious) using silencing RNAs (siRNAs) in HEK293T human embryonic cells. Transfection with pools of four unique siRNAs specifically targeting either *U2AF1*, *HNRNPA0*, *SF3B3*, or *NECAP1* for 72 h achieved robust reductions in the targeted mRNAs (Fig 4A–D). A splice-switching oligonucleotide known to increase SMN2 exon 7 inclusion [SMN2 SSO (d'Ydewalle *et al*, 2017)] was used as a positive control. Changes in SMN1, SMN2-FL, and SMN2Δ7 mRNA expression were assessed with RT–qPCR. Knockdown of U2AF1 increased SMN2-FL mRNA by 50% compared with cells treated with a scrambled control siRNA (Fig 4E and F). Conversely, knockdown of *SF3B3* resulted in a 35% decrease in SMN2-FL and 62% increase in SMN2Δ7 mRNA levels (Fig 4E and F). As expected, there was no effect on inclusion of exon 7 in SMN1 mRNAs (Fig 4G).

*U2AF1* is the top-ranked potential modifier according to *P*-values from step 3. We used the output of step 3 for ranking because of the established relationship between higher full-length/truncated SMN2 transcript ratio and reduced SMA severity (Prior *et al*, 2009; Wu *et al*, 2017). To determine whether the knockdown of the top SMA disease modifier, *U2AF1*, also increases SMN2-FL expression in the SMA disease context, we assessed changes in SMN2 mRNA and

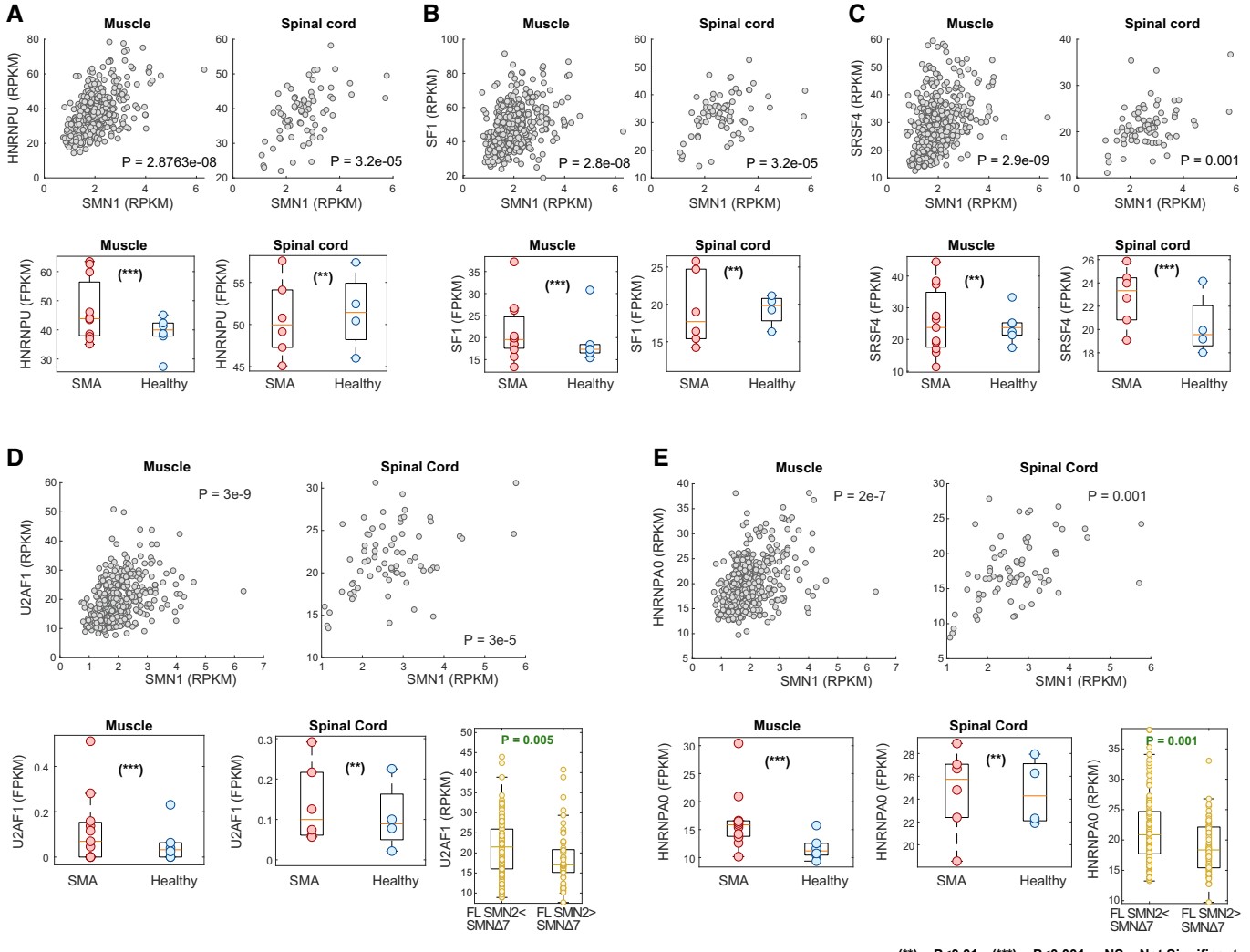

**Figure 3. GENDULF identification of SMA modifier genes.**

A–C Previously reported modifiers. Upper panels: Scatter plots associating the expression of the GCD (SMN1) vs. identified PMs in healthy GTEx muscle tissues (left panels) and spinal cord tissues (right panels); the expression of *SMN1* (*x*-axis) vs. that of (A) *HNRNPU*, (B) *SF1* and (C) *SRSF4*. Bottom panels: Boxplots associating the expression of the identified DPMs in case–control studies; the expression of (A) *HNRNPU*, (B) SF1, and (C) *SRSF4* in SMA and in healthy-control muscle (left panels) and spinal cord (right panels) tissues.

D, E Candidate modifiers. Upper panels: Scatter plots associating the expression of the GCD (SMN1) vs. identified PMs in healthy GTEx muscle tissues (left panels) and spinal cord tissues (right panels); the expression of SMN1 (*x*-axis) vs. that of the negative DPM modifiers (D) *U2AF1* and (E) *HNRNPA0*. Bottom panels: Boxplots associating the expression of the identified DPMs in case–control studies; the expression of (D) *U2AF1* and (E) *HNRNPA0* in SMA and in healthy-control muscle (left panels) and spinal cord (middle panels) tissues. Right panels show the levels of these predicted modifiers in healthy muscle samples with low ratio of SMN2-FL to SMN2Δ7 and those with high ratio of SMN2-FL to SMN2Δ7.

Data information: Empirical *P*-values significance is indicated for two thresholds (**P*-value < 0.01 and ****P*-value < 0.001, using the permutation test defined in the Materials and Methods section). The *P*-values denoted in the upper panels of (A, B, C, D, and E) are for the hypergeometric enrichment test. There are six control (blue) and eleven SMA biological replicates (red) from muscle tissues and four control (blue) and six SMA biological replicates (red) from spinal cord tissues. The Reads Per Kilobase Million (RPKM) measure was used in the GTEx dataset, and the Fragments Per Kilobase Million (FPKM) measure was used in the SMA case–control dataset.

SMN protein expression after treating primary fibroblasts derived from a SMA patient with siRNAs targeting U2AF1. Fibroblasts derived from a healthy carrier for SMA were also treated with scrambled siRNA and were used as a reference. We found that knockdown of *U2AF1* with 25 nM or 100 nM of siRNA for 72 h, resulted in robust reductions of *U2AF1* mRNA and protein expression (Fig 4H, K, and L), was associated with 27 and 46% increases in SMN2-FL mRNA and 59 and 76% increases in SMN protein levels, respectively (Fig 4I–M). Little change was observed in SMN2Δ7 mRNA levels, raising the possibility that *U2AF1* may function at the splicing and transcriptional levels. Together, these data verify that reduction of *U2AF1* expression in the SMA genetic background is associated with increased SMN2-FL mRNA and SMN protein expression.

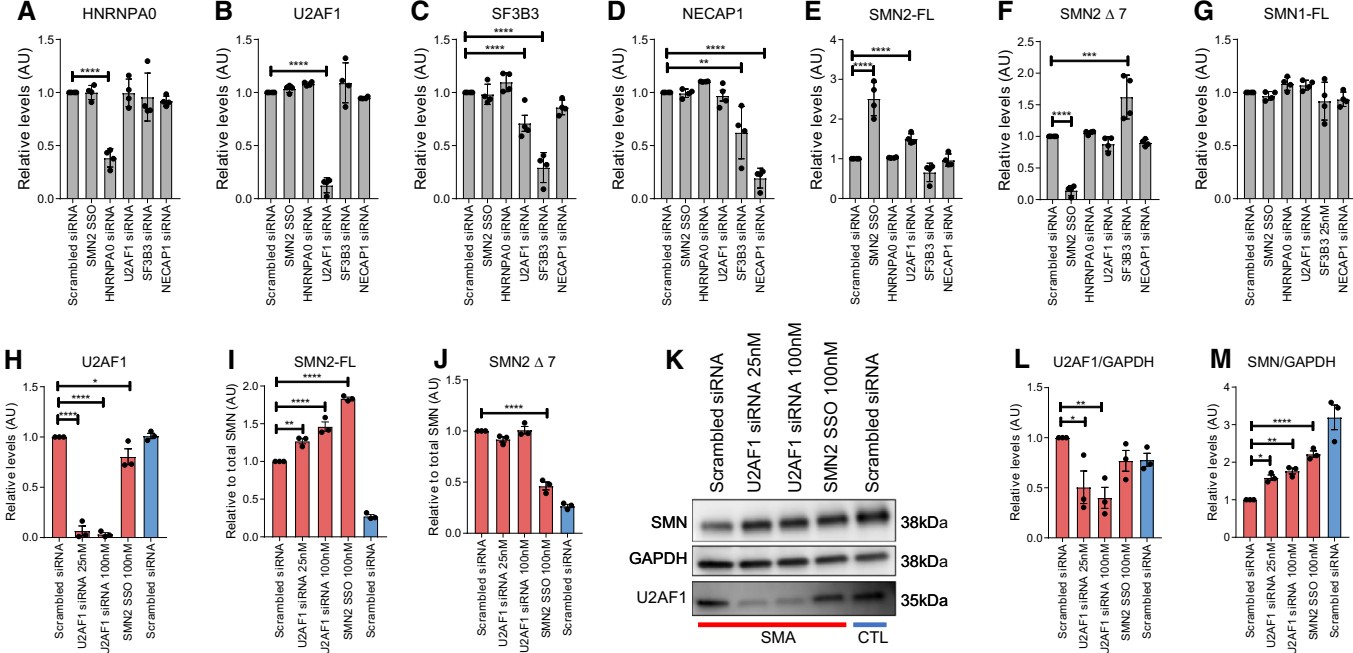

**Figure 4. Knockdown of U2AF1 enhances SMN expression.**

A–D    Expression of (A) *HNRNPA0*, (B) *U2AF1*, (C) *SF3B3*, or (D) *NECAP1* 72 h post-transfection with either SMN2 SSO or knockdown of each target using siRNAs in HEK293T cells.

E–G    Expression of (E) SMN2-FL, (F) SMN2Δ7, or (G) SMN1-FL following *SMN2* SSO or siRNA transfection (*n* = 4).

H–J    mRNA expression of (H) *U2AF1*, (I) SMN2-FL, or (J) SMN2Δ7 in SMA patient-derived fibroblasts (red bars) 72 h post-transfection with SMN2 SSO, scrambled siRNA, or *U2AF1* siRNA at the indicated dose. Unaffected, SMA carrier fibroblasts (blue bars) treated with scrambled siRNA.

K    Representative Western blot.

L, M    Quantification of protein expression of (L) *U2AF1* or (M) SMN normalized to GAPDH (*n* = 3).

Data information: *P < 0.05; **P < 0.01; ***P < 0.001; ****P < 0.0001; empirical *P*-values are determined by a simulation and sampling method described in Materials and Methods. The error bars show the Standard Error of Mean (SEM). Data from (A–G) represent 4 biological replicates. Data from (H–M) represent 3 biological replicates. Three technical replicates were performed during qPCR for each biological replicate and averaged. Scrambled siRNA condition for each experiment was set to 1.

# Discussion

While there is growing recognition that genomic context strongly modulates the clinical severity of many monogenic disorders, few modifier genes have been identified to date (Kousi & Katsanis, 2015). Recently, Kerner *et al* (2020) proposed a case-only, disease-specific, genome-wide strategy based on DNA variants identified in whole exome sequencing. One can also search simultaneously for modifiers of many disease genes in large sequencing databases such as ExAC by identifying putatively healthy individuals who carry one or more variants that are expected to cause a monogenic disorder (Tarailo-Graovac *et al*, 2017); this approach also uses DNA analysis (not gene expression) and is not targeted to a disease since databases such as ExAC ascertain healthy individuals.

Although existing strategies to identify monogenic disorder modulators, including association and linkage studies and genome-wide screens, have successfully identified modifiers of disease, methods that incorporate gene expression data could help uncover novel modifiers that might compensate at the level of transcriptional regulation. There are existing methods that use GTEx to identify variants associated with various traits and diseases. Several

approaches have been established to integrate GWAS and GTEx gene expression data (Giambartolomei *et al*, 2014; Gamazon *et al*, 2015; Gusev *et al*, 2016; Aguet *et al*, 2017), but these are limited to associations with common traits that vary among the GTEx samples, which exclude monogenic disorders. More recently, at least three methods that integrate GWAS and GTEx data have been developed and applied to analyze some specific GWAS data searching for modifiers of CF (Hormozdiari *et al*, 2016; Wen *et al*, 2017; Gong *et al*, 2019). One strength of these methods is that they can analyze individual SNPs and one weakness is that they require large GWAS studies for modifiers, which are feasible for very few Mendelian disorders. Even more recently, a new method called ANEVA has been described to combine GTEx data with DNA sequencing to interpret whether point mutations are the primary cause of a rare Mendelian disease in individual patients (Mohammadi *et al*, 2019). To follow up on GWAS studies that find multiple individual SNPs associated with a complex disease, there are methods to seek statistically significant interactions between pairs of SNPs to strengthen the combined association (e.g., Yip *et al*, 2018 and references therein). However, these methods assume that there are many SNPs that are individually significantly associated with the disease and thus are less suitable for monogenic disorders that have one primary

causal gene plus modifiers. In principle, direct computations of expression quantitative trait loci (eQTL) could be used to find DNA variants in PMs that affect the expression of the GCD in the tissue(s) of interest. However, human eQTL studies including GTEx (The GTEx Consortium, 2015, 2017) are positionally biased because they find mostly eQTLs in which the variant is close to the target gene (called "cis-eQTLs") rather than trans-eQTLs, for which the variant and the target gene may be far apart or on different chromosomes. Adding to this existing collection of methods, GENDULF allows a positionally unbiased search for PMs irrespective of gene location or proximity to the GCD.

GENDULF can be used to further analyze previously identified genomic loci in which the modifier gene has not been distinguished. It can be particularly useful when biological mechanisms underlying the disease are understood. Here, we demonstrated the utility of GENDULF in two relatively common monogenetic diseases, CF and SMA, because both have been studied to search for modifiers and patient gene expression data are available. Like GWAS, the second step of GENDULF uses a case–control design and the output items (SNPs for GWAS, genes for GENDULF) are ranked by significance. However, taking a different tack than GWAS, GENDULF uses two different sources of data for step 1 and step 2. This enables it to remove from consideration most genes at step 1, before analyzing any patient data at step 2. In this way, GENDULF provides a data-driven filtering step that can be incorporated with other bioinformatic analysis tools or wet laboratory experiments. GENDULF operates at the level of genes rather than at the level of SNPs and analyzes gene expression data to assess how changes in transcription may contribute to the underlying mechanism of modifier genes. If both gene expression and genotyping data are available, then GENDULF and GWAS can be used in series (and in either order) such that one method identifies candidates and the other method evaluates the candidates.

GENDULF is based on the assumption that for a subset of monogenic disorders, modifier genes are inactivated in healthy tissues when a disease-causing gene is downregulated or lost, compensating for its low levels. Going beyond conventional approaches that look at the differential expression of genes, GENDULF examines how *the relationship of expression levels between pairs of genes* changes in healthy vs. disease states. The output of GENDULF is a list of candidate modifiers which is enriched with true modifiers that may be then used for experimental screens. Alternatively, GENDULF may be used in combination with other approaches, as demonstrated here by combining association studies with GENDULF for CF. GENDULF may also be augmented by incorporating a third, diseases-specific or knowledge-based step, as exemplified for SMA, to yield a manageable list of candidates of interest for small-scale experiments. In addition to identifying modifiers of CF from whole genome transcriptomic data, GENDULF also successfully identified the known CF modifier genes contained within multi-genic loci previously associated with CF severity. At four of five of these loci, the genes assigned with the lowest *P*-values from GENDULF coincided with those reported in the literature (Wright *et al*, 2011; Corvol *et al*, 2015). When applied to SMA, the added third disease-relevant step provided a refined list of modifiers, which were then tested and validated in cultured cells from SMA patients.

Like CF, SMA is a monogenic disorder with a range of clinical phenotypes. Several modifier genes have been reported to modulate SMA downstream of SMN deficiency including plastin 3, neurocalcin delta, and calcineurin-like EF-hand protein 1 (Riessland *et al*, 2017; Janzen *et al*, 2018). Perhaps, the most critical determinant of SMA severity is the amount of SMN2 exon 7 retention. Enhancing exon 7 inclusion is a principal objective of SMA therapeutics. Nusinersen is an antisense oligonucleotide now in commercial use that targets an intronic splicing silencer N1 (ISS-N1) in intron 7 of SMN2 sterically hindering the binding of HNRNPA1 and thus promoting exon 7 inclusion (Hua *et al*, 2011; Finkel *et al*, 2016). Small molecule splice modifiers are currently in clinical trials and may promote exon 7 inclusion by binding the 5′ splice site of exon 7 and stabilizing U1 snRNP interactions (Palacino *et al*, 2015; Sivaramakrishnan *et al*, 2017). While these novel therapeutics are improving outcomes in SMA patients, clinical and biochemical responses are variable (Sumner & Crawford, 2018; Ramos *et al*, 2019), raising the real possibility that modifier genes play important roles not only in initial SMA disease severity, but also in therapeutic responsiveness.

When applied to SMA, GENDULF identified a number of possible modifiers known to be involved in pre-mRNA splicing. The top predicted modifier, *U2AF1*, is a known component of the spliceosome and important for 3′ splice site selection (Lee & Rio, 2015). While *U2AF1* is classically understood to promote 3′ splice site recognition, there is evidence linking *U2AF1* depletion to increased exon inclusion (Kim *et al*, 2018). The reasons for this are poorly understood, but in the context of *SMN2* may involve reduced recognition of the 3′ splice site of exon 8, thus increasing the probability of 3′ splice site recognition at exon 7, as speculated by (Jodelka *et al*, 2010). Another consideration is that HNRNPA1 binds an exon silencer site (ESS) that overlaps with the *U2AF1*-binding site at the 3′ splice site of exon 7. As it has been demonstrated that recruitment of HNRNPA1 can be U2AF1-dependent (Tavanez *et al*, 2012), it is possible that *U2AF1* disruption reduces recruitment of HNRNPA1, thus promoting exon 7 inclusion (Koed Doktor *et al*, 2011). Regardless of the mechanism, our experimental data confirm that knockdown of *U2AF1* is associated with increased *SMN2*-FL mRNA and SMN protein expression in patient-derived fibroblasts. These data are in agreement with prior data from cultured HeLa cells co-transfected with a *SMN2*-splicing reporter and an siRNA-targeting *U2AF1* (Xiao *et al*, 2012). Importantly, it has also been shown that the disruptions of *U2AF1* expression that occur in patients with lung adenocarcinoma also cause changes in *SMN2* splicing (Kim *et al*, 2018), emphasizing that variations of *U2AF*1 and *SMN2* splicing also occur in humans *in vivo*. Future studies will be needed to determine how variation of *U2AF1* specifically in SMA patients affects disease severity as well as response to splice-modifying therapeutics. In addition to *U2AF1*, the identification of *SF3B3* as a modifier whose downregulation results in a decrease in SMN2-FL and increase in SMN2Δ7 mRNA levels together imply that GENDULF may also be used to uncover targets whose activation can improve disease phenotype.

Like many computational genomics methods, GENDULF analyzes large-scale, noisy omics data and has a few notable limitations; first, it can only be applied to monogenic diseases in which low GCD expression is a possible disease mechanism, as Step 1 of GENDULF relies on analyzing low GCD expression samples from healthy individuals in GTEx. Second, GENDULF can only be applied to diseases in which there is considerable phenotypic variability.

Third, GENDULF has only been applied to disorders with autosomal recessive inheritance in our study. GENDULF could be extended to X-linked recessive inheritance for which one must take gender into account in both steps. Extensions to dominant inheritance disorders would require evidence of haploinsufficiency rather than a gain of function mutation. However, even in haploinsufficiency-driven dominant diseases, sufficiently low expression of the GCD by definition would be expected to result in the disease phenotype and would hence not likely be observed in healthy individuals. A fourth limitation is that GENDULF requires transcriptomics data in both GTEx (step 1) and derived from disease and control samples (step 2). The case–control data are often of modest size; the tissue that is most representative of the disease may be missing in GTEx; some genes or transcripts may have the expression value as 0 for many samples in GTEx (as we observed for SMN2-specific transcripts); some genes may not have expression measured at all in the case–control samples (as we observed for two previously published CF lung modifier genes). Moreover, the available transcriptomic data from case–control studies do not always perfectly match the tissue identity of the healthy tissues available from GTEx, the cell types collected in GTEx may not represent that most affected in disease, and different RNA sequencing preparation methods may introduce additional confounding factors. It is important to note that the sensitivity of GENDULF is generally low, even if substantially higher than random. Incorporating a disease-specific step into the framework of GENDULF, as done in the SMA analysis, increased the sensitivity of the predicted sets and yielded a manageable list of strong candidates. This should be done whenever possible. Regardless, GENDULF predictions should be followed by experiments to test the emerging candidates and validate predictions.

Addressing the latter topic, exploration of the literature revealed several diseases that might be amenable to GENDULF. The relevant literature includes instances of a GCD with a modifier, instances of true digenic inheritance where neither gene alone is sufficient to cause the full disease, and other more complicated two-gene scenarios. When there are sufficiently many examples, these three scenarios can be formally distinguished with machine learning techniques (Gazzo *et al*, 2017; Versbraegen *et al*, 2019). We identified seven sets of diseases that have so far received considerable attention in such studies: CF, SMA, globinopathies (thalassemias, sickle-cell anemia), deafness, long QT syndrome, ciliopathies especially Bardet–Biedl syndrome, and hypogonadotropic hypogonadism (including Kallman Syndrome) (Schäffer, 2013; Kousi & Katsanis,

2015; Gazzo *et al*, 2016). These disorders should be amenable to the GENDULF method except for deafness because measuring gene expression in the inner ear is challenging. Indeed, previous studies of digenic inheritance found almost exclusively cases in which mutations or reduced expression of the second gene are associated with a more severe phenotype (Gazzo *et al*, 2016), again with the exception of deafness (Yousaf *et al*, 2018). The case of ciliopathies is striking since every one of the many examples in which two genes are implicated together as causing or worsening a human ciliopathy, the deleterious mutation or reduced expression of the second gene is associated with a more severe phenotype. This phenomenon in ciliopathies includes the formally identified disease modifier CCDC28B (Cardenas-Rodriguez *et al*, 2013) as well as many families considered to have digenic inheritance in which the primary gene (here we would call it the GCD) has biallelic mutations and the secondary gene (here we would call it the PM) has a heterozygous mutation. In contrast to the human data, a mouse study shows that mutation of the ciliopathy gene *Mkks* mitigates the severity of *Cep290* deficiency (Rachel *et al*, 2012). Thus, ciliopathies would be good candidates for GENDULF. Additional examples from the literature are provided in an Appendix Discussion.

Although diseases are often referred to as "monogenic", there are multiple genes influencing disease manifestations, patient treatment strategies, and outcomes (Dipple & McCabe, 2000). Both *CFTR* mutations and the splicing variant in *SMN2* have the property that the active protein expression in patients is low, but not completely absent. Hence, it is possible to find healthy individuals who have gene expression levels comparable to patients; it remains to be determined whether GENDULF could work for more extreme cases in which the expression of the GCD in patients is completely absent. By using GENDULF, the gray area between a monogenic disorder diagnosis and disease presentation can begin to become elucidated. With the growing number of publicly available transcriptomics resources, GENDULF offers new and innovative ways to study gene expression. Because alternative splicing and post-transcriptional modification of many genes can alter protein expression in the absence of changes at the RNA level, future studies may further expand the scope of GENDULF to assess changes of disease genes at the proteome and phosphoproteome levels, which have not been included in this analysis. Importantly, GENDULF can be readily modified to incorporate such additional data types, allowing for the discovery of novel modifiers that may function at the post-transcriptional level.

# Materials and Methods

### Reagents and Tools table

| Reagents/Resource | Reference or Source | Identifier |
|---|---|---|
| **Software** | | |
| GENDULF | This study https://github.com/noamaus/GENDULF | |
| **Gene expression datasets** | | |
| Healthy gene expression data | The GTEx Consortium (2013) | |
| CF lung case–control | Wright *et al* (2006) | GSE2395 |
| CF colon | Stanke *et al* (2014) | GSE15568 |

**Reagents and Tools table**  (continued)

| Reagents/Resource | Reference or Source | Identifier |
|---|---|---|
| SMA muscle | This study | GSE159642 |
| SMA spinal cord | This study | GSE159642 |
| Curated modifiers list | Dataset EV2 | |
| **Transcript distinction** | | |
| TruSeq RNA protocol | The GTEx Consortium (2013) | |
| **Other** | | |
| siRNA/SSO | Appendix Table S3 | |
| Taqman Probes for qPCR detection | Appendix Tables S4 and S5 | |
| SYBR Green primers for qBase + GeNorm analysis | Appendix Table S6 | |
| Antibodies and dilutions for Western blot | Appendix Table S7 | |

## Methods and Protocols

### The GENDULF pipeline

GENDULF is implemented as a set of Python functions in a single module. As described in Fig 1 and in the following subsections, GENDULF includes two or three major steps.

1 GENDULF Step 1. Identification of PM (potential modifier) genes. The first step aims to identify genes that, when downregulated, may compensate for the low expression of a GCD *in healthy individuals*. This step takes as input only data from healthy individuals in GTEx and the user specifies one or more tissues of interest, such as lung for CF. Using the reads per kilobase of transcript per million mapped reads (RPKM) gene expression values downloaded directly from the GTEx data version V6p, GENDULF identifies genes that have very low expression (in the bottom 10% of expression levels) in healthy individuals in which the GCD is particularly low (in the bottom 10% of expression levels). To evaluate the significance of the overlap between samples with low expression of the candidate and that of the GCD, we use a Bonferroni-corrected hypergeometric *P*-value with $\alpha = 0.01$, i.e., *P*-value $< 0.01/m$ when $m$ is the number of hypotheses tested (genes). We focus on the bottom 10% as in CF, patients with less than 4–20% of normal CFTR expression can show clinical phenotypes (Kerem *et al*, 1997; Ramalho *et al*, 2002). The healthy tissue that is studied is that most affected in the disease; if multiple tissues are affected, the analysis is performed for each affected tissue separately and then the intersection of the genes selected for every tissue is passed to step 2. The output of step 1 is a list of genes that have lower than expected expression in individuals who are in the lowest 10% of expression of the GCD. The 10% threshold can be changed by the user.

2 GENDULF step 2 and the Empirical *P*-value computation. While the first step of GENDULF may identify candidates that are simply co-expressed with the GCD that may or may not have a direct functional association, step 2 of GENDULF is designed to eliminate such PMs. In step 2, GENDULF compares the expression of each PM in an independent (non-GTEx and not preselected to have low expression of the GCD) set of affected cases vs. controls, testing the hypothesis that the PM expression is higher or comparable in the cases compared with the controls.

GENDULF then excludes candidates showing a clear pattern of co-expression with the GCD in both healthy and affected individuals and are hence unlikely to be true modifiers. GENDULF Step 2 thus returns a subset of the PM genes, termed disease-associated PM (DPM) genes, which are more likely to compensate for the loss of the GCD in the disease.

- To compare the expression of the PMs in cases and controls, we use a permutation test rather than a standard test of differential expression. By doing so, step 2 excludes PMs that are low expressing in cases relative to controls, yielding a list of genes whose expression is either higher or comparable between cases and the controls. Because the tissue of origin may differ between the GTEx and the case–control data, we compare the PM ranks between the cases and controls with the PM ranks in the GTEx data between GCD-low and non-GCD-low samples, as explained below.

- The sample size of the case–control studies analyzed in step 2 is typically small, usually an order of magnitude smaller than the sample size available to us in healthy tissues and used in step 1 of GENDULF. In addition, the tissue of origin in the case–control studies may not perfectly match that of GTEx, used for step 1, and the usage of different sources may introduce other confounders and technical covariates. To overcome these challenges, we introduce an empirical *P*-value to quantify the difference in the PM-GCD association between healthy and disease tissues. For this, the following notations are defined: the healthy-control samples used only in step 2 are denoted as $S_c$, whereas the disease samples as $S_D$. The GTEx tissue samples from step 1 where the GCD is low (bottom 10 percentile) are denoted as $S_{GL}$, and the rest of the GTEx samples as $S_{GH}$.

- In this test, the null hypothesis $H_0$ is that the PM expression is always low when the GCD is low, regardless of the tissue type (GTEx healthy tissues of case–control data). Hence, the null hypothesis is that the PM expression is as much lower in $S_D$ compared with $S_c$, as it is lower in $S_{GL}$ compared with $S_{GH}$. The alternative hypothesis $H_A$ is that the PM expression is low when the GCD is low only when the phenotype observed is healthy. Hence, the alternative hypothesis is that the PM expression is not lower in $S_D$ compared with $S_c$ as much as it was lower in $S_{GL}$ compared with $S_{GH}$.

- To perform this test, we randomly sample 10,000 groups of (A) $|S_D|$ samples from the respective GTEx healthy tissue in which the

GCD is low (maintaining the GENDULF defined threshold, of bottom 10% of GCD expression levels) and (B) $|S_c|$ samples where the GCD expression is not low. Within each group, sampling is without replacement. Thus, each group of $|S_D|$ GCD-low expression samples, and $|S_c|$ not GCD-low expression samples has the same sample sizes as the original case–control study. For each group $i$ (=1,…,10,000), we calculate the number of Pairs $P_i$ between from all possible pairs $< x_L, x_H >$ ($x_L$ in $S_{GL}$ and $x_H$ in $S_{GL}$), where $x_L < x_H$. Since there are 10,000 sampled groups, this step gives 10,000 number of pairs $P_i$. We count the number of times, $T$, the number of pairs $< y_D, y_C >$ ( $y_D$ in $S_D$ and $y_C$ in $S_c$) where $y_D < y_C$ is greater or equal to $P_i$. We define the permutation $P$-value to be the ratio of $T$ divided by 10,000, the number of replicates. If this empirical $P$-value is smaller than 0.05 (i.e., $T < 500$), $H_0$ is rejected. Because $T$ is small, the event that the number of pairs is high in the case–control study compared with a sampled group of healthy individuals is rare. Therefore, we infer that the difference in gene expression of the PM in disease vs. control *is consistently of greater magnitude in healthy tissues* with low vs. not low GCD expression.

### Application of GENDULF to CF

For the identification of CF genetic modifiers, data from previously published case–control studies were separately analyzed for lung and colon tissues, as distinct CF phenotypes are associated with each of these tissues:

For lung, the case–control gene expression study used is from nasal brushings of the inferior turbinates of mild and severe CF patients (with identical homozygote $\Delta F508$ mutations) and healthy controls (Wright *et al*, 2006).

- For colon, the case–control gene expression study used is from rectal mucosal epithelia from CF patients (bearing $\Delta F508$ mutation) and healthy controls (Stanke *et al*, 2014). (Appendix Table S1)
- To evaluate an overlap between GENDULF-predicted modifiers and previously verified modifiers collected from the literature (Dataset EV2), we applied the hypergeometric enrichment test. The hypergeometric test is standard in statistics when comparing the overlap between a new set of multiple items to an established set of multiple items (Johnson & Kotz, 1977). The hypergeometric test is widely used in interpreting results of gene expression analysis and GWAS to assess whether the set of genes identified is enriched for any class of genes such as (i) genes already published or (ii) genes in a particular biological pathway such as DNA damage repair (Falcon & Gentleman, 2008). In the standard application of the hypergeometric test in genomics, all genes are weighed equally in the analysis, regardless of how much data there is for each gene or by what ratio the gene expression differs or according to what $P$-value is assigned to the gene (Falcon & Gentleman, 2008).

### Application of GENDULF to SMA and a disease-specific GENDULF step 3

The two steps of GENDULF were applied consecutively to muscle and spinal cord tissues, as a similar SMA phenotype is associated with both of these tissues.

- GENDULF Step 1 was applied to the two tissues (muscle, spinal cord) sequentially, and then, the results are combined via intersection of gene lists during the SMA analysis.
- GENDULF Step 2 was applied to identify genes from step 1 that lose their association with SMN1 in disease tissues based on an empirical $P$-value calculation. RNA sequencing data from SMA and control muscle and thoracic spinal cord tissues were analyzed to identify those genes from step 1 that were not significantly downregulated in the disease tissues vs. healthy controls.
- In the case of SMA, GENDULF adds a third step. In this SMA-specific step, we search for PMs that passed the test at step 2, and whose decreased expression is associated with higher expression of the full-length (exon 7 containing) *SMN2* transcript relative to the truncated SMN2 transcript, as full-length *SMN2* can at least partially compensate for the function of the mutated *SMN1* and can influence disease severity. To this end, we searched in GTEx for genes whose downregulation is associated with a higher ratio of full-length SMN2 vs. exon 7 skipped SMN2 in healthy muscle tissues (as GTEx spinal cord has insufficient number of samples for this evaluation). To this end, we applied a one-sided rank-sum test testing the association of the expression of each PM emerging from step 2 with a high (> 1) ratio of full-length to exon 7 skipped SMN2 expression levels. The third step was used as a criterion to rank the PMs, which revealed *U2AF1* as the top predicted modifier for SMA.

### Power analysis estimating the number of positive and negative samples necessary for GENDULF step 2

Because step 2 requires an independent sample dataset, which may not be available in published data, users of GENDULF may naturally be interested in an estimate of how many cases and controls should be collected to form such a case/control set. Therefore, GENDULF offers a power calculation function to detect which PMs reported in step 1 have a high probability to pass the test in step 2 as a function of the number of cases and controls used in step 2 (thus limiting type 2 error of GENDULF step 2).

- The empirical test receives as input

1    The tissue of interest,
2    A GCD,
3    A PM (found in GENDULF step 1),
4    $N$ = number of positive (case) and negative (control) samples.
     It outputs the level of confidence (likelihood) in which GENDULF step 2 is expected to correctly identify the PM if it is a DPM.

- The procedure applies ($R = 10$) iterations of sampling $N$ samples of the GCD expression from $S_{GL}$ and again another $N$ samples from the same distribution $S_{GL}$, and runs GENDULF step 2 procedure to examine if it correctly accepts this PM as DPM (as the sampled distributions are not the same as the original distribution, which are from $S_{GL}$ and $S_{HL}$). The fraction of times (out of denominator R) that this procedure accepts the PM as DPM is the level of confidence in which GENDULF step 2 is expected to correctly identify the PM if it is a DPM. To make an overall estimate for the minimal number of samples (case and controls) to collect, the minimal

value of $N$ with which the average over the confidence levels for all PMs of a given tissue and GCD (obtained from GENDULF step 1) is above 0.8 is taken. The power calculation for CF in lung tissue is provided in Appendix Fig S2.

- In addition to the standard power analysis that estimates the number of cases and controls that would limit type 2 error, GENDULF can estimate the number of cases and controls that would limit type 1 error, and thereby improve its positive predictive value (PPV). To evaluate the "prospective PPV", we consider a situation in which the null hypothesis should not be rejected; i.e., we want to avoid a false positive that would decrease the PPV. This estimation of prospective PPV differs fundamentally from the calculation of "retrospective PPV" in Appendix Fig S1C because the prospective calculation looks only at hypothetical expression data to determine what are true-positive replicates and false-positive replicates. In contrast, for the retrospective analysis in Appendix Fig S1C, the true positives are the predicted modifiers that have been reported in the literature and the false positives are predicted modifiers that have not been reported in the literature. The primary motivation for new methods such as GENDULF is the general awareness that many modifier genes still remain to be found, given our yet limited knowledge of the biology of many monogenic disorders. Hence, the retrospective PPV for the reported modifiers as shown in Appendix Fig S1C is expected to be low, even if the hypothetical PPV based on simulations may be high.

- As for the power estimation, we sample $N$ cases and $N$ controls. For this type 1 error estimation, we take $N$ samples of the GCD expression from $S_{GL}$ and $N$ samples from $S_{GH}$ and evaluate whether the null hypothesis should be rejected with each sample. We do this sampling for all PMs of a given tissue and GCD (obtained from GENDULF step 1). Then, we compute what is the probability among all PMs that the type 1 error is below 0.05. We found that for any number of samples considered in Appendix Fig S2, the probability is greater than 0.95 that the type 1 error is below 0.05.

### Evaluation of GENDULF for identifying modifiers in candidate chromosomal segments arising from GWAS or linkage studies

In principle, GENDULF can search for PMs across the genome. In practice, it is also of interest to apply GENDULF to selected lists of PMs produced by other methods. GENDULF supports this functionality by allowing the user to specify a list of genes (or all genes) in a genomic interval, which GENDULF can then further rank according to their likelihood to be disease modifiers. For example, GWAS are often used for the identification of genetic modifiers and genetic linkage analysis of pedigrees has also been attempted. However, these studies often reveal a chromosomal segment that may include multiple genes. As a complementary analysis to our primary method of *de novo* modifier prediction, we studied the ability of GENDULF to identify other potential modifiers lying in these regions.

- We separately analyzed five loci implicated in the CF lung phenotype (Corvol *et al*, 2015) and the two loci implicated in housing modifiers of the CF intestinal phenotype (Sun *et al*, 2012). We used the UCSC Genome Browser (https://genome.ucsc.edu) to identify genes closely located within each reported genomic region

and evaluated the $P$-value resulting from applying GENDULF step 1 for each of these genes.

- For the lung phenotype, a similar trend was observed for four of the five loci (Fig 2G), which was not observed for the fifth locus (Appendix Fig S3A).

- For the colon phenotype (meconium ileus), the trend observed was similar, but of lesser strength (Appendix Fig S3B).

- To evaluate the robustness of our analysis, we randomly shuffled the CFTR expression in GTEx healthy lung samples and repeated the GENDULF analysis. For all 10,000 repetitions, we do not find any gene with a significant $P$-value when correcting for multiple hypotheses testing (GENDULF step 1, hence yielding a permutation $P$-value < 1e-4).

### Robustness analysis applied to CF lung disease

We performed a sensitivity, specificity, and positive predictive value (PPV) analysis for the CF modifiers identified for lung tissues (where the largest number of previously reported modifiers is available).

- To evaluate the sensitivity of GENDULF step 2, we sampled sets of patients with similar sizes as the case–control data, in the same manner described for the empirical $P$-value calculation for GENDULF step 2.

- For all measures, we find that the results reported are not too sensitive to small variations in the threshold values used in this step: using 0.15, 0.2, 0.35, and 0.3 yield comparably significant results (Appendix Fig S1). However, lowering the threshold below 0.1 yields far less significant results, likely because the number of samples having this percentile of expression is too small.

- The list of candidates obtained with GENDULF is highly enriched with previously reported modifiers, and the sensitivity is over 2 orders of magnitude higher than random (Appendix Fig S1). To narrow down to a smaller list of modifiers for small-scale experiments, it is recommended to use an additional step, either by incorporating GENDULF with another approach (as demonstrated for CF), or to include a disease-specific knowledge-based step (as demonstrated for SMA), which also allows a ranking of the predictions.

- To assess whether the GENDULF $P$-values are inflated, we did the following computational experiments for CF and lung tissue. First, we tabulated the step 2 $P$-values for all genes with enough lung data in GTEx if all genes pass step 1 regardless of the step 1 $P$-value (Appendix Fig S6A and B). Second, we tabulated the step 2 $P$-values for all interesting genes that pass the $P$-value threshold at GENDULF step 1 (Appendix Fig S6C). 4% of the $P$-values are < 0.01 and 1% of the $P$-values are < 0.001. The rightward shift in the plot of Appendix Fig S6C is expected because it represents the signal of true modifiers as supported by the high specificity in Appendix Fig S1.

### SMA human tissue collection as a precursor to validating results from GENDULF step 3 for SMA

Following legal and institutional ethical regulations, as well as patient- or parent-informed consent, thoracic spinal cord or iliopsoas was dissected from patients at the time of autopsy and immediately flash frozen in liquid nitrogen. Samples were stored at −80°C

until RNA isolation for RNA sequencing experiments. Protocols for autopsy were approved by the Institutional Review Boards at the Johns Hopkins University School of Medicine. Some thoracic spinal cord samples were obtained through the National Institutes of Health NeuroBioBank (University of Maryland, Baltimore, MD) following proper procedures. The human subject experiments conformed to the principles set out in the WMA Declaration of Helsinki and the Department of Health and Human Service Belmont Report. More details about the SMA samples are provided in Appendix Tables S1 and S2.

### SMA RNA isolation and library preparation as precursor to validating results from GENDULF step 3 for SMA

Total RNA was extracted from about 20 mg tissues using RNeasy® Plus Mini Kit (Qiagen, Valencia, CA) according to manufacturer's instructions. The RNA integrity (RIN) was examined using TapeStation 2200 (Agilent, Santa Clara, CA). All the RNA used in RNA sequencing had a RIN of 5.8 or above. The RNA-seq libraries were prepared using TruSeq Stranded Total RNA Library Prep Kit with Ribo-Zero Gold (Illumina, La Jolla, CA) following the manufacturer's instruction. The libraries were pooled for pair-end 50-bp sequencing on HiSeq™ 2500 (Illumina, La Jolla, CA). Paired-end reads were then aligned to hg19 reference sequence using the STAR aligner (Dobin et al, 2013) with default parameters. Gene expression was quantified using the HTSeq package (Anders et al, 2015) with the intersection_nonempty option and the GENCODE v19 annotation (GRCh37) as the reference.

### Identification of SMN1 and SMN2 transcript isoforms as a precursor to validating results from GENDULF step 3 for SMA

SMN1 and SMN2 transcript isoforms were classified in the GTEx data(The GTEx Consortium, 2013). The full-length SMN2 transcript used for identification is ENST00000380743, and the SMN2Δ7 transcript is ENST00000380741. The full-length SMN1 transcript used for identification is ENSG00000172062 (the sum of all SMN1 transcripts). The distinction between these different transcripts was made using long RNA-seq reads with 76-base, paired-end Illumina TruSeq RNA protocol, and unique read mapping that enabled identification of the close splice isoforms with accuracy that could not be obtained with expression arrays (The GTEx Consortium, 2013).

### SMA-derived cell lines and transfections

We obtained human SMA patient-derived fibroblast (Coriell Cell line ID: GM00232; 0 copies SMN1; 2 copies SMN2) and unaffected SMA carrier-derived fibroblast (Coriell Cell line ID: GM08333; 3 copies SMN1; 1 copy SMN2) cell lines from Coriell Cell Repository and HEK293T cells from American Type Culture Collection (ATCC). We maintained HEK293T cells and fibroblasts in Dulbecco's modified Eagle medium supplemented with 10% fetal bovine serum, 50 U/ml penicillin, and 50 μg/ml streptomycin.

HEK293T cells were transfected with siRNAs and splice skipping oligonucleotides (SSOs provided by Ionis Pharmaceuticals) using DharmaFECT 1 (GE Healthcare) using recommended manufacturer's instructions. Fibroblasts were transfected with siRNAs and SSOs using cytofectin (Generously provided by Ionis Pharmaceuticals). siRNAs, SSOs, and lipid reagent were diluted in reduced serum Opti-MEM medium, transfection complexes were added dropwise to the cells, and siRNAs/SSOs were incubated for the indicated time periods before

RNA or protein isolation. Sequence information for SSOs and catalog numbers for siRNAs used are listed in Appendix Table S3.

### RNA extraction and quantitative PCR on SMA samples and controls

RNA was isolated using the RNeasy mini kit (Qiagen) per manufacturer's recommendations including optional on-column DNase treatment. First strand cDNA was synthesized using total RNA and the High Capacity cDNA conversion kit (Thermo Fisher) per manufacturer's recommendations. Quantitative real-time PCR was performed with the HT7900 Real-Time PCR System (Thermo Fisher) using Taqman Universal PCR master mix (Thermo Fisher) or the SYBR Green universal PCR master mix (Thermo Fisher) and the appropriate custom Taqman assays (Thermo Fisher, S6), commercially available Taqman assay (Thermo Fisher, Appendix Table S4), or primers (Appendix Table S5).

- RT–qPCR experiments were quantified using qBase + software (Biogazelle) (Vandesompele et al, 2002). For each experiment, eight reference genes were run. GeNorm analysis was performed to evaluate and select (at least 2 of) the most stable reference genes (Vandesompele et al, 2002).
- Target gene expression levels were normalized to the most stable reference genes and calibrated to a control, and all expression levels are expressed as calibrated normalized relative quantities (CNRQ) (Vandesompele et al, 2002), except where indicated. Sequence information for SYBR Green primers used in GeNorm analysis listed in Expanded View Table S6.

### Western blots for proteins encoded by candidate SMA PMs

RIPA supplemented with protease inhibitors was incubated with samples for 10 min on ice. Samples were kept on ice and sonicated using a rod sonicator with 30× 1-s pulses. Samples were centrifuged at 20,000 g for 15 min at 4°C to clear cellular debris. Protein concentrations were determined using the MicroBCA Protein Assay kit and a SpectraMAX Plus spectrophotometer. Equal amounts of protein were diluted in RIPA buffer, supplemented with sampling buffer (250 mM Tris–HCl pH 6.8, 10% SDS, 30% glycerol, 5% β-mercaptoethanol, and 0.02% bromophenol blue), and denatured for 10 min at 95°C. Protein lysates were resolved on 4–15% Mini-PROTEAN TGX Precast Protein Gels (Biorad). The Trans-Blot Turbo transfer system (Bio-Rad) was used to transfer proteins to PVDF membranes. Membranes were blocked for 60 min at room temperature (RT) in 5% BSA in TBS supplemented with 0.1% Tween-20. Primary antibodies were diluted in 2.5% BSA in TBS supplemented with 0.1% Tween-20 and incubated with the membrane rocking overnight at 4°C. Secondary antibodies conjugated to alkaline phosphatase were diluted 1:5,000 in 2.5% BSA in TBS supplemented with 0.1% Tween-20 and incubated for 60 min at RT. Protein bands were visualized by incubating membranes with enhanced chemifluorescence (ECF) for 5 min at RT and scanned utilizing the LAS4000 (GE Healthcare) equipped with a Cy2 filter. A list of antibodies and dilutions used is available in Appendix Table S7.

### Statistical analyses of in vitro data from SMA cases and controls

Statistical analysis was performed using GraphPad Prism version 7.03. The level of significance was set to 0.05. One-way ANOVA was performed to compare treatments groups to a control group

(scrambled siRNA). For fibroblast experiments, control fibroblast line was not included in statistical analysis. Bonferroni test was used to correct for multiple comparisons in all cases.

## Data availability

The datasets and computer code produced in this study are available in the following databases:

i  Both the datasets and code are publicly and freely available from the GitHub repository: https://github.com/noamaus/GENDULF.
ii The transcriptomic data of the SMA patients analyzed are available from GEO (GSE159642; https://www.ncbi.nlm.nih.gov/geo/query/acc.cgi?acc=GSE159642).

**Expanded View** for this article is available online.

## Acknowledgements

We thank the Cancer Data Science Lab (CDSL) members for helpful discussions and comments. We also thank Giovanni Coppola for helpful discussion and Qing Wang for sequencing library preparation. This research was supported in part by the Intramural Research Program of the National Institutes of Health, National Cancer Institute and National Library of Medicine. This work was also supported in part by NIH (NINDS) grants R01NS096770 and 5F31NS105376.

## Author contributions

Research supervision: ER, AAS, and CJS; Computational approach conception and design: NA and ER; Computational approach development: NA; Experimental procedure design: CJS and DMR; Experiments: DMR; Writing and testing of python package: NA and AAS; Computational analysis assistance: HK. Transcriptomic data collection: IZ and TOC; Manuscript writing: NA, CJS, ER, DMR, and AAS.

## Conflict of interest

The authors declare that they have no conflict of interest.

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
