## [Review Process File · Molecular Systems Biology]

The GENDULF algorithm: mining transcriptomics to uncover modifier genes for monogenic diseases

Noam auslander, Daniel Ramos, Ivette Zelaya, Hiren Karathia, Thomas Crawford, Alejandro Schäffer, Charlotte Sumner, and Eytan Ruppin

DOI: [10.15252/msb.20209701](https://doi.org/10.15252/msb.20209701)

Corresponding author(s): Eytan Ruppin (eyruppin@gmail.com)

Review Timeline:

Submission Date:	11th May 20
Editorial Decision:	19th Jun 20
Revision Received:	24th Jul 20
Editorial Decision:	25th Aug 20
Revision Received:	20th Oct 20
Accepted:	3rd Nov 20

Editor: Maria Polychronidou

Transaction Report:

Thank you again for submitting your work to Molecular Systems Biology. We have now heard back from the three referees who agreed to evaluate your study. Overall, the reviewers recognize that the presented approach seems interesting. However, they raise a series of concerns, which we would ask you to address in a major revision.

I think that the recommendations of the reviewers are rather clear and there is therefore no need to repeat the points listed below. All issues raised by the reviewers would need to be convincingly addressed. Please let me know in case you would like to discuss any of the issues raised.

On a more editorial level, we would ask you to address the following issues.

REFEREE REPORTS

Reviewer #1:

This is a very interesting and well-written paper. I enjoyed reading the manuscript and it was overall very clear. The authors present a tool called GENDULF that can be used to identify modifier genes for known monogenic disorders using gene expression data. They evaluate their work on CF and SMA. GENDULF's contribution, as stated by the authors, is that it can reduce the number of potential modifier genes that need testing, which makes it a useful tool in filtering pipelines. They demonstrate the relevance of the GENDULF approach in the context of CF, where they are able to identify some of the previously known modifier genes, and in the context of SMA, where they identify U2AF1 and three other potential modifiers, which are also evaluated using knock-down experiments. The approach is a statistical in nature and it estimates changes in low-low expression between two genes when moving from healthy to disease related data. The base assumption is that genes downregulated significantly when the GCD is also expressed at low levels are potential modifier candidates, which are further analyzed in a step in the context of the diseased tissue.

Although I think this a very interesting work, I have some concerns.

First, this work aims at identifying epistatic relationships. It would be useful to the reader that this explicitly stated and that the reader is informed about the state-of-the-art of predictive/filtering methods aiming to identify such relationships. It would moreover be useful to know how GENDULF compares to other methods that are trying to achieve the same. As it is written now, the method appears isolated and it is difficult to assess how it moves beyond what can already be achieved via other approaches (e.g. Yip, D. K.-S., Chan, L. L., Pang, I. K., Jiang, W., Tang, N. L. S., Yu, W., & Yip, K. Y. (2018). A network approach to exploring the functional basis of gene-gene epistatic interactions in disease susceptibility. *Bioinformatics*, 34(10), 1741-1749, but there are most likely also others). If possible, some benchmarking/comparison should be done.

Second, and this is a more important concern, I would like to see a sensitivity/specificity analysis of the predictions made by this approach (provide also the confusion matrix in SI). When considering the excels listing the known CF modifiers and the results of step 1 and step 2, the sensitivity appears to be low, this makes me also wonder about the specificity. Which known modifiers disappear after the first and second step?

Clearly, if this GENDULF approach would be used for detection of modifiers, using potentially some form of ranking, then all CF modifiers should be in the top, whereas those that have no epistatic relation should disappear (you speak also about top DPMs), but I didn't see how you rank them, or I might have misread something. Some order statistics could maybe be provided. Information should also be provided about the number of erroneous modifiers that are identified, and potentially how to reduce them.

Third, as the approach is statistical, it would be useful to know what the limits are one the sample size needed (power analysis) to make relevant predictions. Can this be assessed? I assume this is how you arrived at 10000 but this should be clearly shown using for instance a power analysis and not just put as an adhoc value. It may even work with smaller sample sizes.

Some minor concerns :

On line 193-194, you make a strong statement. Can you provide support for that? Otherwise I would remove it.

As the paper is well-written I have only one minor comment on the Figure 1 (top part). Why do you show 3 non-PM cases? Is it just to illustrate the difference between PM and non-PM? Then 2 plots are enough?

Reviewer #2:

The identification of modifier genes/variation is an important issue in monogenic disorders. Actually, there are not *stricto sensu* monogenic disorders since the severity of the phenotypic consequences vary even in subjects with the same molecular defect. Thus any advance in identifying modifiers is welcome both in understanding the molecular pathophysiology and in therapeutic options.

This present study uses gene expression differences to identify negative modifiers. The rational is well explained and the data presented are encouraging.

Comments

1. The method presented is suitable for autosomal recessive disorders and probably not for

dominant. Wondering if the method could be applied to dominant Loss of Function variants, and in that case the method could detect the expression of the other (non-mutant but variable-expression) allele that could be a strong modifier.

2. This reviewer would wish to see the example of the sickle cell disease (and/or beta thalassemia) included in the analysis. The HBB-related disorders are the easiest ones to study. Expression in blood, common, with known modifiers. I am surprised that the authors did not present HBB-related disorders as the prime example.

3. What is the false positive rate? Were there any known modifiers in CF or SMA that the method did not identify? Furthermore, what is the false positive rate?

4. I think the method presented is useful because it provides a list of candidate genes; the magnitude of that list likely depends on the thresholds that the authors use.

5. There was not all a discussion about eQTLs. Gene expression is related to eQTLs and the GTEx (and many other) projects have provided a wealth of eQTLs tissue-specific or not. The eQTLs could be detected by DNA sequence and thus there is no need of a specific tissue. This reviewer would wish to see a comparison between eQTL and RNAseq prediction of modifiers.

6. Could the authors provide some indication of the sample size of step 2? Obviously the method would be under-powered in the majority of recessive disorders that are more rare than CF, SMA and HBS.

Reviewer #3:

Auslander et al. describe a novel approach, GENDULF, for finding modifier genes for Mendelian diseases by coexpression analysis. Their approach takes advantage of existing large cohorts of healthy individuals and smaller gene expression analyses in case-control settings to form hypotheses of genes whose expression may modify Mendelian disease severity. The hypothesis is nice and biologically sound. In this paper, they test their method in CF and apply it to SMA, with results that are interesting but (in my understanding) mostly refining previous findings of modifiers rather than providing truly novel modifiers. The main text is well written, and altogether I rather like this work. However, there are several weaknesses that need to be addressed in revisions. A key weakness is that in a manuscript that is about a new method, there should be more material testing the assumptions, robustness, and thresholds of this method to ensure that it works as it should, and to guide a potential user. There are a lot of nonstandard statistical or analytical approaches that are not very well justified, and based on the data provided, I'm not sure that the different steps and tests are well calibrated under the null. Some of the steps needing more detail are listed below, together with other concerns in approximate order of importance.

- There is insufficient consideration and discussion of the diverse sources of gene expression variation. It is now known that in human tissue samples, cell type composition and its variation between individuals is the key source of variation in gene expression levels including coexpression patterns. On top of that, GTEx has diverse post-mortem effects and other complex technical covariates. I think it's quite likely that most of the genes picked up by Step 1 of GANDULF are simply genes expressed in the same cell type as the GCD. It would be interesting to see what the results would look like if covariates (GTEx shared these, I think) were corrected, but at the very minimum this needs to be discussed.

- Most of the main text is well written, but Methods altogether and some parts of methods-type text elsewhere need additional work to improve structure and clarity. I struggled to understand some of the key steps, and the Methods overall is structured in a way that makes it difficult to find all relevant info. It needs more work.

- It was a real struggle to understand initially what actually happens in step 1, and I'm still not sure. I

assume that in Figure 1, each dot is an individual. If I understand correctly, the algorithm doesn't actually analyze e.g. a correlation of a GCD and another gene, but rather takes the lowest 10% GCD-expressing individuals and tests differential expression in other genes for these individuals versus others. I don't think that a scatterplot in Figure 1 communicates this well. But maybe I'm mistaken. And if it tests for differential expression, why is this done with a hypergeometric test and not e.g. with DESeq or another standard method?

- Step 2, Page 24, rows 539-550: I don't understand what is being said in this text. Are there some copy-paste errors? "more significant than the change of expression" - of what? Why is sampling from GTEx being referred to as S_D and S_C when above it's defined that these are disease and control samples? I can't really review this since at this moment I don't understand what the test does, but as far as I understand, it makes an assumption that under the null, disease and control individuals differ from each other no more or less than how GTEx S_GL and S_GH samples differ from each other. I'm not sure this is a valid assumption, and more detail should be provided that this test is well calibrated under the null. Again, a comparison to a standard differential expression test would be very much needed to ensure that the somewhat nonstandard approach taken here actually works.

- Step 2: Since the evidence here is absence of differential expression, power is a key question. There should be more material evaluating what kind of power is needed in this step, and how differences e.g. in sample size affects the results.

- I couldn't locate supplementary tables 1-3, so I couldn't review these. Related to these, "we find that the GENDULF-predicted DPMs are highly enriched with modifiers of CF manifestations in the lung" - enriched compared to what? I hope that the enrichment considers only tested genes that have ~equal power (matched for expression level if needed). For such a small number of overlaps the p-value seems extremely good, perhaps too good. Also, why the hypergeometric test; it seems like an unusual choice?

- The results of how the varying thresholds (10% vs something else) affect the results should be shown as a supplementary figure. Also, I don't understand this: "However, lowering the threshold towards 0 risks having a too small a sample size for step 2." Wouldn't this be "too small a number of genes for step 2"?

- Discussion: "Therefore, neither GWAS nor GENDULF can fully distinguish between association and causality" - This need to be revised or justified better. Genetic variation does not suffer from the same reverse causation and confounder issues as gene expression analysis.

- Discussion: "However, in contrast with GWAS, GENDULF filters out the vast majority of genes at step 1." GWAS does filter out the vast majority of loci and typically yields a shorter gene list than GENDULF. Please rephrase.

- Step 3, Figure 3D, E and S2C: Looking at the ratio of the exon 7 including transcript and its association to the modifiers, why is the transcript ratio made binary? It could be studied as a continuous variable. Also, if the hypothesis is that the modifier affects the exon inclusion, the current way is analyzing it a bit backwards, isn't it? Also, it's not clear to me why this is a third sequential step instead of having Steps 1 and 2 looking at transcript ratios.

- SMA analysis, Step 3: "Thus, in Step 3 of the GENDULF analysis we set to identify, among the candidates identified from GENDULF steps 1-2, those that are most significantly associated with SMN2 exon 7 inclusion, pinpointing U2AF1 and HNRNPA0 (Figure 3D-E)." According to the supplementary table, these are not the only or most significant genes. The full results and why you ended up analyzing these two in detail need to be reported.

- In the siRNA and RT-qPCR experiments, how many biological and technical replicates were used for siRNA knockdowns and for RT-qPCR? The variation between replicates in Fig 4 looks extremely small for these kinds of experiments.

- What's the GTEx version used?

- Fig 2D: Please add a label (individuals, right?) for the rows. The legend is also quite unclear.

- Figure 1: It would be good to have "expression" in the axis labels.

Dear Editor and Reviewers,

This document is the point-by-point response to reviewers for the revision of manuscript MSB-20-9701 entitled “The GENDULF algorithm: mining transcriptomics to uncover modifier genes for monogenic diseases”; the cover letter to the Editor is a separate document.

Reviewer comments are copied below verbatim and interspersed with our responses. We took the liberty of numbering the concerns of reviewers 1 and 3, which were transmitted originally without numbers. In the case of reviewer 3, the paragraph to which we would logically assign the number 1 is split into two parts, which we designate 1a and 1b. When comments from different reviewers overlap, we note the overlap and give equivalent responses, so that the response to each reviewer is self-contained.

To aid the reviewers in finding various changes, we highlight the changes to the main text in yellow and we indicate either the subsections or the pages where changes were made; page numbers refer to the numbers at the bottom center of each page with tracked changes off and assume that the title page is page 1. Please note that by generating a pdf for the reviewers, the *Molecular Systems Biology* manuscript management system may shift the page numbers and that having tracked changes on may also shift page numbers.

Reviewer #1:

This is a very interesting and well-written paper. I enjoyed reading the manuscript and it was overall very clear. The authors present a tool called GENDULF that can be used to identify modifiers genes for known monogenic disorders using gene expression data. They evaluate their work on CF and SMA. GENDULF's contribution, as stated by the authors, is that it can reduce the number of potential modifier genes that need testing, which makes it a useful tool in filtering pipelines. They demonstrate the relevance of the GENDULF approach in the context of CF, where they are able to identify some of the previously known modifier genes, and in the context of SMA, where they identify U2AF1 and three other potential modifiers, which are also

evaluated using knock-down experiments. The approach is a statistical in nature and it estimates changes in low-low expression between two genes when moving from healthy to disease related data. The base assumption is that genes downregulated significantly when the GCD is also expressed at low levels are potential modifier candidates, which are further analyzed in a step in the context of the diseased tissue.

Response: We thank the reviewer for the kind words and for the accurate summary of our manuscript.

Although I think this a very interesting work, I have some concerns.

1. First, this work aims at identifying epistatic relationships. It would be useful to the reader that this explicitly stated and that the reader is informed about the state-of-the-art of predictive/filtering methods aiming to identify such relationships. It would moreover be useful to know how GENDULF compares to other methods that are trying to achieve the same. As it is written now, the method appears isolated and it is difficult to assess how it moves beyond what can already be achieved via other approaches (e.g. Yip, D. K.-S., Chan, L. L., Pang, I. K., Jiang, W., Tang, N. L. S., Yu, W., & Yip, K. Y. (2018). A network approach to exploring the functional basis of gene-gene epistatic interactions in disease susceptibility. *Bioinformatics*, 34(10), 1741-1749, but there are most likely also others). If possible, some benchmarking/comparison should be done.

Response: We agree with the reviewer that it should be explicitly stated that GENDULF aims to identify epistatic interactions and we have edited the Introduction accordingly [page 4]:

“The interactions we seek between the GCD and modifiers are akin to some definitions of the genetic term “epistasis”, but we avoid using this term because it is sometimes associated with formal measures of overall organism fitness, which we do not compute.”

We also accept the reviewer's suggestion to mention the paper by Yip et al. in the Discussion, although the topic is somewhat different. This study examines pairs of SNPs associated with a complex disorder whose associations are not independent in a statistically significant way. We added the following to the first paragraph of the Discussion where we also discuss other methods related to GWAS [page 20]:

“To follow-up on GWAS studies that find multiple individual SNPs associated with a complex disease, there are methods to seek statistically significant interactions between pairs of SNPs to strengthen the combined association (e.g. Yip *et al.* 2018 and references therein). However, these methods assume that there are many SNPs that are individually significantly associated with the disease and thus are less suitable for monogenic disorders that have one primary causal gene plus modifiers.”

More generally, during our literature review, we have previously identified very few methods that aim to identify modifier genes for monogenic disorders. Nonetheless, in the course of preparing this revision, we updated our literature search to look for recent and even more distantly related papers. We did not find any new papers about methods that are comparable to GENDULF. However, we did add a note relating to the method of (M. Tarailo-Graovac et al. *Genet. Med.* 2017; 19:1300-1308), explaining why it is not directly comparable to GENDULF [page 20].

During our updated literature review, we did identify a new, comprehensive review of cystic fibrosis modifiers (S. Shanthikumar et al. “Gene modifiers of cystic fibrosis lung disease: A systematic review”, *Pediatric Pulmonology* 2019; 54: 1356-1366), which we now cite. Importantly, this review highlights that that two other previously identified modifiers of CF: *KRT8* and *MUC1* were also identified as modifiers by the GENDULF algorithm (Table EV1).

2. Second, and this is a more important concern, I would like to see a sensitivity/specificity analysis of the predictions made by this approach (provide also the confusion matrix in SI). When considering the excels listing the known CF modifiers and the results of step 1 and step 2,

the sensitivity appears to be low, this makes me also wonder about the specificity. Which known modifiers disappear after the first and second step?

Response: We thank all three reviewers for similar comments requesting data on sensitivity and specificity (see also reviewer 2, comment #3 and reviewer 3, comment #6). We added an analysis showing the sensitivity and specificity for GENDULF steps 1 and 2 applied to CF lung disease (for which there are more known modifiers than for other diseases/tissues examined and hence the sensitivity and specificity estimations are based on this prior knowledge), with different thresholds for step 1.

While the specificity is very high (especially when using the 0.1 threshold), the reviewer is correct in that the sensitivity is not very high, but it is higher than random by >200 fold. We added a description of this analysis to the Methods subsection entitled “Robustness analysis applied to CF” [page 31], and the results are shown in Expanded View Figure EV2, which is now cross-referenced in Results [page 10].

In addition, we provide below tables reporting the previously verified CF modifiers for lung and colon tissues that were identified by GENDULF steps 1 and 2. In colon, all previously verified CF modifiers picked up by GENDULF step 1 are retained in GENDULF step 2. In lung CF, however, two of the previously verified CF modifiers that were picked up by GENDULF step 1 (*SFTPA1*, *SLC26A9*) disappear in step 2, but that is simply because these two genes were not measured in the CF lung case-control dataset.

Lung		
step 1 gene	step 1 P-value	Step 2 pass
AGTR2	3.99E-10	Pass
EHF	2.18E-11	Pass
GSTP1	8.23E-07	Pass
KRT8	9.85E-13	Pass
SCNNIA	6.07E-09	Pass

SFTPA1	3.99E-10	Not measured in the data (GSE2395)
SFTPA2	6.07E-09	Pass
SFTPD	7.73E-08	Pass
SLC26A9	3.67E-14	Not measured in the data (GSE2395)
SLC6A14	3.67E-14	Pass

Colon		
step 1 gene	step 1 P-value	Step 2 pass
CLCA1	1.79E-11	Pass
FABP1	3.96E-14	Pass
MUC2	4.86E-07	Pass
SLC4A4	4.10E-09	Pass

We summarized the findings in these Tables in the Results subsection entitled “Applying GENDULF to identify gene modifiers of CF”, but did not include the Tables in the revised manuscript (as these are contained in the Appendix data).

3. Clearly, if this GENDULF approach would be used for detection of modifiers, using potentially some form of ranking, then all CF modifiers should be in the top, whereas those that have no epistatic relation should disappear (you speak also about top DPMs), but I didn't see how you rank them, or I might have misread something. Some order statistics could maybe be provided. Information should also be provided about the number of erroneous modifiers that are identified, and potentially how to reduce them.

Response:

In general, genes are ranked by p-value at the end of step 2 (as in our CF analysis) or at the end of step 3 (as in our SMA analysis). As we note in the previous response and in the response to reviewer #2, comment #2, two related limitations of GENDULF are that i) the expression of the PM has to be measured in the fitting tissue in sufficiently many GTEx samples and with non-

zero values and ii) the expression of the PM has to be measured in the case-control sample in order for GENDULF to have any chance to find a PM.

GENDULF may also be used in conjunction with one or more of the following procedures to reduce the number of false positives: (a) incorporation of another approach, as exemplified for the case of CF and GWAS predicted modifiers. (b) Incorporation of a third step, searching for a disease-specific attribute (such as exemplified for SMA with the third step applied), or (c) performing knowledge-based filtering - such as filtering genes based on the biological pathways and knowledge of their possible relevance. Any of these complementary methods would substantially reduce the number of false positives and narrow down the candidates to a smaller list that is more likely to contain relevant modifiers. We specifically exemplify this for cases (a) in CF and (b) in SMA, which yield a small list of modifiers that could be tested experimentally. To rank the genes in SMA, we used this additional step 3 based on full-length/truncated SMN2 transcript ratios, as this has previously been shown to modify disease severity.

In our sensitivity/specificity analysis we have taken a conservative approach, assuming that the collection of modifiers reported in the literature is complete. However, we note that there may well be yet unknown modifiers and possibly, also known modifiers that were missed by our collection. Given that the collection is not perfect, and that many modifiers have not yet been discovered or reported in the literature, there is no way to know if a predicted modifier is erroneous, or simply not yet found. The steps to take in order to narrow-down the list of predicted modifiers or rank them based on a knowledge-based step are now discussed the second paragraph of the Discussion [page 21].

We added a description of the sensitivity/specificity analysis to the Methods subsection entitled “Robustness analysis applied to CF” [pages 31-32], and the results are shown in Expanded View Figure EV2, which is now cross-referenced in Results [page 10].

4. Third, as the approach is statistical, it would be useful to know what the limits are one the sample size needed (power analysis) to make relevant predictions. Can this be assessed? I assume this is how you arrived at 10000 but this should be clearly shown using for instance a

power analysis and not just put as an ad hoc value. It may even work with smaller sample sizes.

Response: We thank all three reviewers for similar comments requesting a power analysis (see also reviewer #2, comment #6; reviewer #3, comment #4). We have now added a power estimation function to the GENDULF code, estimating the number of cases and controls necessary for a given GCD and tissue type. The power calculation is based on the interim results of step 1, which uses GTEx and does not depend on disease-specific sample collection; the latter are then used as a basis for estimating the number of samples that one needs to have a desired level of power at step 2 of GENDULF.

Given a specific GCD and a tissue of interest, we apply GENDULF step 1 to obtain a list of candidate modifiers (PMs). The power analysis then uses these PMs to estimate a minimal number (N) of samples that should be analyzed in step 2, such that with N case and N control samples there would be over 0.8 (default level, can be modified by the user) confidence in correctly identifying a DPM within the list of given PMs. The full and formal description of this analysis is provided in the new Methods section entitled “Power analysis estimating the number of positive and negative samples necessary for GENDULF step 2 “. The results of this analysis for CF lung disease are presented in Expanded View Figure EV3 and we added some text to Results summarizing the power estimation for CF lung tissues and the default threshold of 0.8 are on page 10 of the revised manuscript.

The number 10,000 that the reviewer asks about is the number of replicates in the GENDULD step 2 empirical p-value calculation. The number of replicates is unrelated to the number of samples needed to achieve a desired level of power in step 2.

Some minor concerns:

5. On line 193-194, you make a strong statement. Can you provide support for that? Otherwise I would remove it.

Response: We removed the statement entirely. The reviewer is correct that it was too speculative.

6. As the paper is well-written I have only one minor comment on the Figure 1 (top part). Why do you show 3 non-PM cases? Is it just to illustrate the difference between PM and non-PM? Then 2 plots are enough?

Response: We thank the reviewer for this very helpful comment; reviewer #3 comment #2 is an expanded version of the same comment. We agree that the upper part of Figure 1 was unnecessarily confusing, and the legend was too terse. We have simplified the upper part of this figure to show two schematic scatter plots that illustrate the significant association between a PM and the GCD on the left, and the lack of significant association between a non-PM and the GCD on the right. We now added the following text to the legend:

“GENDULF does not compute a correlation across the whole range of expression values, but specifically searches only for a significant association at the lower level range, as shown in the region boxed in the left scatter plot, where the dots are in blue and the y-axis is labeled ‘PM’ for ‘potential modifier’. The scatter plot on the upper right depicts an example of relationship between expression of the GCD and another gene in which GENDULF is not expected to find the other gene as a modifier, and hence the y-axis is labeled non-PM expression.”

Reviewer #2:

The identification of modifier genes/variation is an important issue in monogenic disorders. Actually, there are not stricto sensu monogenic disorders since the severity of the phenotypic consequences vary even in subjects with the same molecular defect. Thus any advance in identifying modifiers is welcome both in understanding the molecular pathophysiology and in therapeutic options.

This present study uses gene expression differences to identify negative modifiers. The rationale is well explained and the data presented are encouraging.

Response: We thank the reviewer for these positive and encouraging remarks. We agree that there are not many methods for finding modifiers of (near) monogenic disorders and made a similar comment in response to reviewer #1 comment #1. As described there, we added the following to the first paragraph of the Discussion where we also discuss other methods related to GWAS:

“To follow-up on GWAS studies that find multiple individual SNPs associated with a complex disease, there are methods to seek statistically significant interactions between pairs of SNPs to strengthen the combined association (e.g. Yip *et al.* 2018 and references therein). However, these methods assume that there are many SNPs that are individually significantly associated with the disease and thus are less suitable for monogenic disorders that have one primary causal gene plus modifiers.”

We also appreciate that there are not *stricto sensu* monogenic disorders and we added a comment accordingly in the Introduction [boundary of pages 3-4]:

“We use the term ‘monogenic disease’ for disorders in which mutations in one gene determine who is affected with high penetrance, but variations in that gene alone may not fully explain the variable phenotypes seen in different patients.”

Comments

1. The method presented is suitable for autosomal recessive disorders and probably not for dominant. Wondering if the method could be applied to dominant Loss of Function variants, and in that case the method could detect the expression of the other (non-mutant but variable-expression) allele that could be a strong modifier.

Response: This is a good point for which we thank the reviewer. We now added to the Discussion [page 23]:

“...GENDULF has only been applied to disorders with autosomal recessive inheritance in our study. GENDULF could be extended to X-linked recessive inheritance for which one must take gender into account in both steps. Extensions to dominant inheritance disorders would require evidence of haploinsufficiency rather than a gain of function mutation. However, even in haploinsufficiency-driven dominant diseases, sufficiently low expression of the GCD by definition would be expected to result in the disease phenotype and would hence not likely be observed in healthy individuals.”

2. This reviewer would wish to see the example of the sickle cell disease (and/or beta thalassemia) included in the analysis. The HBB-related disorders are the easiest ones to study. Expression in blood, common, with known modifiers. I am surprised that the authors did not present HBB-related disorders as the prime example.

Response: This is an excellent idea, in principle. The technical difficulty is that sickle cell disease (SCD) is a disorder of erythrocytes, which have no transcription. Since erythrocytes receive their protein contents from megakaryocyte precursors, the ideal “tissue” or “cell type” for GENDULF step 1 would be megakaryocytes, but GTEx does not collect expression data from megakaryocytes. In the absence of a more appropriate data set, we performed GENDULF step 1 using “blood” as the GTEx tissue. We were able to identify a suitable SCD data set for GENDULF step 2 (GEO data set GSE72999). The candidate modifiers for SCD obtained with GENDULF step1+2 are in the following table. The list is shorter than that obtained for CF or SMA, because in GTEx blood tissues, many genes are frequently assigned an expression value of zero (in more than 50% of samples). The genes with zero values of expression in too many samples were excluded from the analysis, as it is unclear how to handle a 0.1 percentile of expression in such cases.

The intersection of the GENDULF predicted modifiers with previously reported modifiers for HBB diseases is noted in the table; the levels of prior evidence for different genes in the list vary, but we distinguished between some evidence (one reference given) and no evidence (blank in the second column). While these are encouraging, we think it is better not to

include these data in the manuscript because of the lack of the appropriate dataset with which to perform GENDULF step 1 and because the manuscript is already quite long.

STEP1+2 (SC disease, Blood, GSE72999)	PubMed ID of one reference reporting this gene in HBB context
AHSP	https://pubmed.ncbi.nlm.nih.gov/16019463/
ALAS2	https://pubmed.ncbi.nlm.nih.gov/31475864
ANK1	https://pubmed.ncbi.nlm.nih.gov/25724378
BAG1	
GLRX5	
GYPB	https://pubmed.ncbi.nlm.nih.gov/31134759
HBA2	https://pubmed.ncbi.nlm.nih.gov/31698466
HBM	https://pubmed.ncbi.nlm.nih.gov/31134759
HBQ1	
KLF1	https://pubmed.ncbi.nlm.nih.gov/20676099
KLHDC8A	
MX11	
NFIX	
NT5M	
OR2W3	
PDZK1IP1	
PHOSPHO1	
PLEK2	https://pubmed.ncbi.nlm.nih.gov/25724378
POLL	
RUNDC3A	https://pubmed.ncbi.nlm.nih.gov/31134759
SEC14L4	
SLC4A1	https://pubmed.ncbi.nlm.nih.gov/25724378
SLC6A8	
STRADB	https://pubmed.ncbi.nlm.nih.gov/25724378
TFR2	https://pubmed.ncbi.nlm.nih.gov/30209118
TNS1	
TRIM10	https://pubmed.ncbi.nlm.nih.gov/31134759
UBA52	

3. What is the false positive rate? Were there any known modifiers of CF or SMA that the method did not identify? Furthermore, what is the false positive rate?

Response: We thank all three reviewers for requesting an analysis of sensitivity and specificity (see also reviewer 1, comment #2 and reviewer 3, comment #6). We added an analysis showing the sensitivity and specificity for GENDULF steps 1 and 2 applied to CF lung disease (for which there are more known modifiers than for other diseases/tissues examined and hence the sensitivity and specificity are based on prior knowledge), with different thresholds for step 1. While the specificity is very high (especially when using the 0.1 threshold), the sensitivity is not very high, but it is higher than random by >200 fold. We added a description of this analysis to the Methods subsection entitled “Robustness analysis applied to CF” [pages 31-32], and the results are shown in Expanded View Figure EV2, which is now cross-referenced in Results [page 10].

4. I think the method presented is useful because it provides a list of candidate genes; the magnitude of that list likely depends on the thresholds that the authors use.

Response: Indeed. Besides the thresholds, there is also the consideration of whether one requires significant results for the same gene in more than one tissue, as we did for spinal muscular atrophy. Hence, the sensitivity and specificity analysis are provided for a range of thresholds for GENDULF step 1. Please see Expanded View Figure EV2.

5. There was not all a discussion about eQTLs. Gene expression is related to eQTLs and the GTEX (and many other) projects have provided a wealth of eQTLs tissue-specific or not. The eQTLs could be detected by DNA sequence and thus there is no need of a specific tissue. This reviewer would wish to see a comparison between eQTL and RNAseq prediction of modifiers.

Response: This is a good point, in principle, but an eQTL approach is difficult to compare to GENDULF in practice. In general, GENDULF seeks modifier genes anywhere in the genome. Our analysis of CF showed some preference for PMs on chromosome arm 7q (location of *CFTR*)

and our analysis of SMA showed some preference for PMs on chromosome arm 5q (location of *SMN1* and *SMN2*). Despite these preferences, the vast majority of PMs are on chromosome arms different from the GCD.

While we seek modifiers anywhere in the genome the first major round of GTEx (GTEx consortium 2015) reported only eQTLs in cis with the target gene and the second major round of GTEx (GTEx consortium 2017) reported less than 5% of its eQTLs with the marker on a different chromosome from the target gene. Other eQTL studies (e.g., Joehanes et al., *Genome Biology* 2017; 18: 16) have been more successful at finding such trans-eQTLs, but in all such studies we have seen, the large majority of eQTLs have the variant placed close to or inside the target gene or at least on the same chromosome arm. To address the reviewer's comment in the manuscript, we now added to the Discussion [page 20]:

“In principle, direct computations of expression quantitative trait loci (eQTL) could be used to find DNA variants in PMs that affect the expression of the GCD in the tissue(s) of interest. However, human eQTL studies including GTEx (GTEx Consortium 2015, 2017) are positionally biased because they find mostly eQTLs in which the variant is close to the target gene (called “cis-eQTLs”) rather than trans-eQTLs, for which the variant and the target gene may be far apart or on different chromosomes. Adding to this existing collection of methods, GENDULF allows a positionally unbiased search for PMs irrespective of gene location or proximity to the GCD.”

6. Could the authors provide some indication of the sample size of step 2? Obviously the method would be under-powered in the majority of recessive disorders that are more rare than CF, SMA and HBS.

Response: We appreciate that all three reviewers asked for a power analysis (See also reviewer #1, comment #3; reviewer #3, comment #4). We have now added a power estimation function to the GENDULF code, estimating the number of cases and controls necessary for a given GCD and tissue type. The power calculation is based on the interim results of step 1, which uses GTEx

and does not depend on disease-specific sample collection; the latter are then used as a basis for estimating the number of samples that one needs to have a desired level of power at step 2 of GENDULF.

Given a specific GCD and a tissue of interest, we apply GENDULF step 1 to obtain a list of candidate modifiers (PMs). The power analysis then uses these PMs to estimate a minimal number (N) of samples that should be analyzed in step 2, such that with N case and N control samples there would be over 0.8 (default level, can be modified by the user) confidence in correctly identifying a DPM within the list of given PMs. The full and formal description of this analysis is provided in the new Methods section entitled “Power analysis estimating the number of positive and negative samples necessary for GENDULF step 2” [page 30]. The results of this analysis for CF lung disease are presented in Expanded View Figure EV3 and we added some text to Results summarizing the power estimation for CF lung tissues and the default threshold of 0.8 [page 10].

Reviewer #3:

Auslander et al. describe a novel approach, GENDULF, for finding modifier genes for Mendelian diseases by coexpression analysis. Their approach takes advantage of existing large cohorts of healthy individuals and smaller gene expression analyses in case-control settings to form hypotheses of genes whose expression may modify Mendelian disease severity. The hypothesis is nice and biologically sound. In this paper, they test their method in CF and apply it to SMA, with results that are interesting but (in my understanding) mostly refining previous findings of modifiers rather than providing truly novel modifiers.

Response: We thank the reviewer for these positive comments. Referring to the last sentence, we emphasize here that *U2AF1* is a novel modifier of SMN expression in the context of SMA, which we also examined experimentally.

The main text is well written, and altogether I rather like this work. However, there are several weaknesses that need to be addressed in revisions. A key weakness is that in a manuscript that is

about a new method, there should be more material testing the assumptions, robustness, and thresholds of this method to ensure that it works as it should, and to guide a potential user. There are a lot of nonstandard statistical or analytical approaches that are not very well justified, and based on the data provided, I'm not sure that the different steps and tests are well calibrated under the null. Some of the steps needing more detail are listed below, together with other concerns in approximate order of importance.

Response: We appreciate the reviewer's specific concerns and have responded to each of them below. To simplify cross-referencing between similar comments from different reviewers, we took the liberty of numbering the reviewer's concerns, and have split the first comment into two parts 1a and 1b.

1a. There is insufficient consideration and discussion of the diverse sources of gene expression variation. It is now known that in human tissue samples, cell type composition and its variation between individuals is the key source of variation in gene expression levels including coexpression patterns. On top of that, GTEx has diverse post-mortem effects and other complex technical covariates. I think it's quite likely that most of the genes picked up by Step 1 of GANDULF are simply genes expressed in the same cell type as the GCD. It would be interesting to see what the results would look like if covariates (GTEx shared these, I think) were corrected, but at the very minimum this needs to be discussed.

Response: We thank the referee for this comment. We recognize two important questions to address in response to this comment.

- (1) Possible issues resulting from diverse sources of gene expression used between the first and second steps of GENDULF. We acknowledge that indeed, the variations between the source of the healthy tissue expression in GENDULF step 1, and that of the case and controls tissues used for GENDULF step 2 may be a confounder for standard statistical analysis that use the distribution of the gene expression. We hence do not directly compare the distribution of a PM across different sources, but we generate an empirical test that compares the ranks of the PM between case vs. control, and healthy tissues with low-GCD vs. those with non-low-GCD. By doing so, we limit the effect of possible

variations resulting from different tissue sources and that introduced by different technical covariates across different studies utilized. We improved the explanation The Methods subsection “GENDULF step 2 and the Empirical P-value computation” [page 26].

- (2) The reviewer mentions that cell type composition and its variation between individuals is the key source of variation in gene expression levels including co-expression patterns. While we agree with the reviewer’s concern that cell type composition may be a confounder to any analysis, including enrichment or co-expression as applied in the first step of GENDULF, we would like to emphasize that the second step of GENDULF is aimed at filtering out PM inferred due to such confounders. If indeed, a gene is picked up by GENDULF step 1 solely because of co-expression in the same cell types with the GCD, it is almost guaranteed to be filtered out in step 2. This is because in such case, we would observe a similar PM-GCD co-expression patterns in step 2 (between cases and controls), as a result of the cell type compositions. This is true for any simple co-expression patterns that may be inferred in GENDULF step 1, which step 2 is aimed at removing. This is now explicitly explained and discussed in the revised Methods section [page 26]:

“While the first step of GENDULF may identify candidates that are simply co-expressed with the GCD that may or may not have a direct functional association, step 2 of GENDULF is designed to eliminate such PMs. In step 2, GENDULF compares the expression of each PM in an independent (non-GTEX and not pre-selected to have low expression of the GCD) set of affected cases vs. controls, testing the hypothesis that the PM expression is higher or comparable in the cases compared to the controls. GENDULF then excludes candidates showing a clear pattern of co-expression with the GCD in both healthy and affected individuals and are hence unlikely to be true modifiers.”

1b. Most of the main text is well written, but Methods altogether and some parts of methods-type text elsewhere need additional work to improve structure and clarity. I struggled to understand some of the key steps, and the Methods overall is structured in a way that makes it difficult to find all relevant info. It needs more work.

Response: We addressed this comment in four ways, the first of which overlaps with responses to other comments.

1. First, in response to other comments from all three reviewers we added Methods subsections entitled “Power analysis estimating the number of positive and negative samples necessary for GENDULF step 2” and “Robustness analysis applied to CF”. We thank the reviewers for pointing out that the information in these two subsections was missing from the initial submission.
2. Second, we changed methods subsections and their order, with more careful consideration of which subsections would interest which types of readers.
3. Third, we changed the headings of several subsections to make them more informative; one specific motivation for the heading changes was to make explicit the logical and chronological connections between the bioinformatics analysis of SMA with GENDULF and the wet lab experiments to validate the GENDULF SMA predictions of modifier genes; the general motivation for changing the headings was to address literally the comment that [the subsection structure in the initial submission] “makes it difficult to find all the relevant info”.
4. Fourth and finally, we did substantial rewriting at sentence level to improve the clarity.

We imagine readers with five different interests, not mutually exclusive, who might focus on different parts of Methods. Readers who are interested in the bioinformatics of GENDULF as a method are advised to focus attention on the first four subsections and the later subsection entitled “Robustness analysis applied to CF lung disease”. Readers who are interested in using GENDULF for a new study in which samples would need to be collected should read the first four subsections and the subsection “Power analysis estimating the number of positive and negative samples necessary for GENDULF step 2”. Readers who are especially interested in CF or who wish to apply GENDULF to some other disease on which there are published data on modifiers should

focus on the subsections entitled “Application of GENDULF to CF”, “Evaluation of GENDULF for identifying modifiers in candidate chromosomal segments arising from GWAS or linkage studies” and “Robustness analysis applied to CF lung disease”. Readers who are especially interested in SMA should read the first four subsections, especially “Application of GENDULF to SMA and a disease-specific GENDULF step 3” and then may wish to skip to the final seven subsections: “SMA human tissue collection as a precursor to validating results from GENDULF step 3 for SMA”, “SMA RNA isolation and library preparation as precursor to validating results from GENDULF step 3 for SMA”, “Identification of *SMN1* and *SMN2* transcript isoforms as a precursor to validating results from GENDULF step 3 for SMA”, “RNA Extraction and quantitative PCR on SMA samples and controls”, “Western blots for proteins encoded by candidate SMA PMs”, and “Statistical analyses of *in vitro* data from SMA cases and controls”.

Since the changes we made to address this comment are large in both number and scale and are inter-dependent, it is not possible to copy short passages from the revised text to show the changes we made.

2. It was a real struggle to understand initially what actually happens in step 1, and I'm still not sure. I assume that in Figure 1, each dot is an individual. If I understand correctly, the algorithm doesn't actually analyze e.g. a correlation of a GCD and another gene, but rather takes the lowest 10% GCD-expressing individuals and tests differential expression in other genes for these individuals versus others. I don't think that a scatterplot in Figure 1 communicates this well. But maybe I'm mistaken. And if it tests for differential expression, why is this done with a hypergeometric test and not e.g. with DESeq or another standard method?

Response: We thank the reviewer for this comment, which overlaps, with reviewer #1 comment #5. We think that the scatterplots are useful because scatterplots are a standard data visualization technique when one is at an early stage of data exploration, perhaps not knowing what one is seeking. Nevertheless, we reduced the number of scatterplots from four to two – one exemplifying the gene expression data pattern that GENDULF seeks and one exemplifying a different type of expression pattern. We agree with reviewers #1 and #3 that the legend of the

upper part of Figure 1 (where the scatter plots are found) was too terse. We now added to the legend:

“GENDULF does not compute a correlation across the whole range of expression values, but specifically searches only for a significant association at the lower level range, as shown in the region boxed in the left scatter plot, where the dots are in blue and the y-axis is labeled ‘PM’ for ‘potential modifier’. The scatter plot on the upper right depicts an example of relationship between expression of the GCD and another gene in which GENDULF is not expected to find the other gene as a modifier, and hence the y-axis is labeled non-PM expression.

The reviewer did understand correctly that the algorithm does not actually analyze a correlation, but the point of Figure 1 is that a preliminary look at the correlation of the expression of the GCD and another gene can give a lot of intuition as to whether GENDULF could find that gene as a potential modifier. GENDULF searches for an enrichment (of the samples with bottom percentiles of expression) between two genes, rather than differential expression of these, which is why hypergeometric enrichment test is a more fitting choice than DESeq or differential expression analyses. We added some intuition about these issues in the new opening subsection of Methods alluded to in the response to comment 1b.

3. Step 2, Page 24, rows 539-550: I don't understand what is being said in this text. Are there some copy-paste errors? "more significant than the change of expression" - of what? Why is sampling from GTEx being referred to as S_D and S_C when above it's defined that these are disease and control samples? I can't really review this since at this moment I don't understand what the test does, but as far as I understand, it makes an assumption that under the null, disease and control individuals differ from each other no more or less than how GTEx S_GL and S_GH samples differ from each other. I'm not sure this is a valid assumption, and more detail should be provided that this test is well calibrated under the null. Again, a comparison to a standard differential expression test would be very much needed to ensure that the somewhat nonstandard approach taken here actually works.

Response: We thank the reviewer for this helpful comment. We now added more text to explain the notation, and we edited the text explaining the null and alternative hypotheses. The S_D and S_C refer the case and control samples for the specific disease that are used only at step 2.

The new text reads:

“For this, the following notations are defined: the healthy-control samples used only in step 2 are denoted as S_C , whereas the disease samples as S_D . The GTEx tissue samples from step 1 where the GCD is low (bottom 10 percentile) are denoted as S_{GL} , and the rest of the GTEx samples as S_{GH} .

In this test, the null hypothesis H_0 is that the PM expression is always low when the GCD is low, regardless of the tissue type (GTEx healthy tissues of case-control data). Hence the null hypothesis is that the PM expression is as much lower in S_D compared to S_C , as it is lower in S_{GL} compared to S_{GH} . The alternative hypothesis H_A is that the PM expression is low when the GCD is low only when the phenotype observed is healthy. Hence, the alternative hypothesis is that the PM expression is not lower in S_D compared to S_C as much as it was lower in S_{GL} compared to S_{GH} .”

GENDULF does not search for genes that are differentially expressed in disease cases vs. unaffected controls. The modifiers we look for may not be differentially expressed at all in disease (cases) vs. controls. We look for genes that are differentially associated with the GCD between healthy and diseases tissues. In the modifiers we look for, we see a strong association in healthy tissues (which may be linked to a modifying action), and in contrast, in disease tissues we do not see this association (as there likely is not much of a modifying action). This does not mean that there is any differential expression of the PM between healthy and controls. The statistical question we addressed is “is the PM and GCD association in the healthy tissues broken in the disease tissues?”. The test we developed for this uses the ranks, rather than expression levels of fold change, which are not comparable between GTEx and the case-control studies, since the gene expression would typically be measured on different experimental platforms for the two samples. The pertaining text now reads [page 32]:

“To assess whether the GENDULF p-values are inflated, we did the following computational experiments for CF and lung tissue. First, we tabulated the step 2 p-values for all genes with enough lung data in GTEx if all genes pass step 1 regardless of the step 1 p-value (Expanded View Figure EV6A,B). Second, we tabulated the step 2 p-values for all interesting genes that pass the p-value threshold at GENDULF step 1 (Expanded View Figure EV6C). 4% of the p-values are < 0.01 and 1% of the p-values are < 0.001 . The rightward shift in the plot of Figure EV6C is expected because it represents the signal of true modifiers as supported by the high specificity in Expanded View Figure EV2.”

4. Step 2: Since the evidence here is absence of differential expression, power is a key question. There should be more material evaluating what kind of power is needed in this step, and how differences e.g. in sample size affects the results.

Response: We appreciate that all three reviewers asked for a power analysis (See also reviewer #1, comment #3; reviewer #2, comment #6). We have now added a power estimation function to the GENDULF code, estimating the number of cases and controls necessary for a given GCD and tissue type. The power calculation is based on the interim results of step 1, which uses GTEx and does not depend on disease-specific sample collection; the latter are then used as a basis for estimating the number of samples that one needs to have a desired level of power at step 2 of GENDULF.

Given a specific GCD and a tissue of interest, we apply GENDULF step 1 to obtain a list of candidate modifiers (PMs). The power analysis then uses these PMs to estimate a minimal number (N) of samples that should be analyzed in step 2, such that with N case and N control samples there would be over 0.8 (default level, can be modified by the user) confidence in correctly identifying a DPM within the list of given PMs. The full and formal description of this analysis is provided in the new Methods section entitled “Power analysis estimating the number of positive and negative samples necessary for GENDULF step 2” [page 30]. The results of this analysis for CF lung disease are presented in Expanded View Figure EV3 and we added some

text to Results summarizing the power estimation for CF lung tissues and the default threshold of 0.8 [page 10].

5. I couldn't locate supplementary tables 1-3, so I couldn't review these. Related to these, "we find that the GENDULF-predicted DPMs are highly enriched with modifiers of CF manifestations in the lung" - enriched compared to what? I hope that the enrichment considers only tested genes that have ~equal power (matched for expression level if needed). For such a small number of overlaps the p-value seems extremely good, perhaps too good. Also, why the hypergeometric test; it seems like an unusual choice?

Response: We regret that, but we note that these Expanded View (also known as Supplementary) Tables were already provided in full in the original submission, so there may have been a mishap in transmission of the files to the reviewers. If you somehow cannot locate these or any other files, please ask for assistance from the *Molecular Systems Biology* Editorial office because we have provided all of them. We have asked the Editor to ensure that all files we submitted in this revision should please be readily accessible for the reviewers.

As to the comment about overlaps, indeed, the enrichment is with experimentally verified modifiers. We now rephrased this to "highly enriched with previously verified modifiers" [page 9]

Along with comment 1b above, the reviewer has a point here that the hypergeometric test may not be familiar to some readers interested in our manuscript. Therefore, we added to the manuscript the two book references cited below, and added to the revised methods section the following text, to support and explain the usage of the hypergeometric test to evaluate the significance of enrichment [pages 28-29]:

"To evaluate an overlap between GENDULF-predicted modifiers and previously verified modifiers collected from the literature (Appendix Dataset EV2), we applied the hypergeometric enrichment test. The hypergeometric test is standard in statistics when comparing the overlap between a new set of multiple

items to an established set of multiple items (Johnson & Kotz, 1977). The hypergeometric test is widely used in interpreting results of gene expression analysis and GWAS to assess whether the set of genes identified is enriched for any class of genes such as i) genes already published or ii) genes in a particular biological pathway such as DNA damage repair (Falcon & Gentleman, 2008). In the standard application of the hypergeometric test in genomics all genes are weighed equally in the analysis, regardless of how much data there is for each gene or by what ratio the gene expression differs or according to what P-value is assigned to the gene (Falcon & Gentleman, 2008). “

6. The results of how the varying thresholds (10% vs something else) affect the results should be shown as a supplementary figure. Also, I don't understand this: "However, lowering the threshold towards 0 risks having a too small a sample size for step 2." Wouldn't this be "too small a number of genes for step 2"?

Response: We thank all three reviewers for requesting an analysis of sensitivity and specificity (see also reviewer 1, comment #2 and reviewer 2, comment #3). We added a sensitivity and specificity analysis as currently described in the response to reviewer 1, comment #2. We added an analysis showing the sensitivity and specificity for GENDULF steps 1 and 2 applied to CF lung disease (for which there are more known modifiers than for other diseases/tissues examined and hence the sensitivity and specificity are based on more prior knowledge), with different thresholds for step 1. While the specificity is very high (especially when using the 0.1 threshold), the sensitivity is not very high, but it is higher than random in >200 fold. We added a description of this analysis to the Methods subsection entitled “Robustness analysis applied to CF” [pages 31-32], and the results are shown in Expanded View Figure EV2, which is now cross-referenced in Results [page 10].

We also rephrased the sentence about using less than 0.1 percentile expression, explaining that it would yield less significant results, probably because the number of samples having this percentile of expression is too small [page 31]:

“However, lowering the threshold towards below 0.1 yields far less significant results, likely because the number of samples having this percentile of expression is too small.”

7. Discussion: "Therefore, neither GWAS nor GENDULF can fully distinguish between association and causality" - This need to be revised or justified better. Genetic variation does not suffer from the same reverse causation and confounder issues as gene expression analysis.

Response: We agree with the referee. We removed the sentence mentioned.

8. Discussion: "However, in contrast with GWAS, GENDULF filters out the vast majority of genes at step 1." GWAS does filter out the vast majority of loci and typically yields a shorter gene list than GENDULF. Please rephrase.

Response: We understand and appreciate this comment and thank the reviewer. We now rewrote the sentence to read [page 21]

“However, taking a different tack than GWAS, GENDULF uses two different sources of data for step 1 and step 2. This enables it to remove from consideration most genes at step 1, before analyzing any patient data at step 2.”

9. Step 3, Figure 3D, E and S2C: Looking at the ratio of the exon 7 including transcript and its association to the modifiers, why is the transcript ratio made binary? It could be studied as a continuous variable. Also, if the hypothesis is that the modifier affects the exon inclusion, the current way is analyzing it a bit backwards, isn't it? Also, it's not clear to me why this is a third sequential step instead of having Steps 1 and 2 looking at transcript ratios.

Response: The ratio of SMN2 transcript expression is made “binary” (larger than/smaller than) because a ratio between two transcripts is not the ideal continuous variable to capture the underlying biology. Especially because there is a proportion of ‘zero’ values for both of these transcripts (10% of the samples have zero expression reported for delta-exon-7 SMN2 and 15%

of the samples have zero expression reported for full length SMN2), which may be a technical artifact rather than real absence. Such zero values may make a very large or very small ratio, which is not proportional, especially when it may only be an artifact (and we cannot determine whether it is real or an artifact). These values would create a relatively high variance to the distribution of the ratios, which may not be biologically correct. Using the “binary” (larger than/smaller than) minimizes the effect that such zero values have on the results. In addition, since the association between exon inclusion and severity of disease is established, it is a fitting variable to use for splitting of the data. However, to address this comment further, we added the coefficients of the rank correlation between the transcripts ratio and the expression of the predicted modifiers (Expanded View Dataset EV3), which mostly yield similar results to the analysis used previously.

To answer the last sentence, step 1 and step 2 are looking at the GCD (SMN1), rather than the ratio (of SMN2 transcripts), which is itself a modifier and not the GCD. In the use of GENDULF in general (and in particular in the CF case) no information about known modifiers or mechanisms is used in GENDULF. The SMA analysis is a special case in which additional information about the known modifier gene, *SMN2*, is available due to published biological knowledge and hence an additional special step 3 is designed to exploit that. A key aspect of the engineering design of GENDULF is that the steps are modular and structured in such a way that disease-specific knowledge can be incorporated after step 2 without changing the software for steps 1 and 2.

10. SMA analysis, Step 3: : " Thus, in Step 3 of the GENDULF analysis we set to identify, among the candidates identified from GENDULF steps 1-2, those that are most significantly associated with SMN2 exon 7 inclusion, pinpointing U2AF1 and HNRNPA0 (Figure 3D-E)." According to the supplementary table, these are not the only or most significant genes. The full results and why you ended up analyzing these two in detail need to be reported.

Response: Thanks. *U2AF1* is indeed the top modifier considering all steps, as step 3 is the critical step used for ranking. *HNRNPA0* was obtained by lowering the strictness of GENDULF to find genes in addition to *U2AF1* that are significantly associated with exon 7 inclusion. To

clarify this point, we added the following text in the subsection entitled “Experimental validation of predicted SMA DPMS” [page 18]

“*U2AF1* is the top-ranked potential modifier according to p-values from step 3. We used the output of step 3 for ranking because of the established relationship between higher full-length/truncated SMN2 transcript ratio and reduced SMA severity.”

11. In the siRNA and RT-qPCR experiments, how many biological and technical replicates were used for siRNA knockdowns and for RT-qPCR? The variation between replicates in Fig 4 looks extremely small for these kinds of experiments.

Response: We thank the reviewer for this comment. We added the following text to the caption of Figure 4 to clarify the issue in hand:

“Data from A-G represent 4 biological replicates. Data from H-M represent 3 biological replicates. Three technical replicates were performed during qPCR for each biological replicate and averaged. Scrambled siRNA condition for each experiment was set to 1.”

12. What's the GTEx version used?

Response: We thank the reviewer for noting this omission. Next to the first occurrence of “GTEx” in Methods, we added the version number V6p.

13. Fig 2D: Please add a label (individuals, right?) for the rows. The legend is also quite unclear.

Response: This is a very good suggestion. We edited the legend to mention that these are indeed individuals.

14. Figure 1: It would be good to have "expression" in the axis labels.

Response: This is a very good suggestion, thanks. We added “expression” in the axis labels of Figure 1 in addition to the more substantial changes we did to Figure 1 and its legend to address reviewer #1 comment #5 and reviewer #3 comment #2.

Thank you for sending us your revised manuscript. We have now heard back from the two reviewers who were asked to evaluate your study. Overall, the reviewers think that the study has improved as a result of the performed revisions. However, as you will see below, they still list a few remaining concerns, which we would ask you to address in a revision.

On a more editorial level, we would ask you to address the following issues.

REFEREE REPORTS

Reviewer #1:

I thank the authors for providing answers to resolve my earlier concerns. Below some remaining remarks/concerns.

You mention an update of the literature review in your answer to my question concerning epistasis prediction. I'm therefore surprised you did not include the following references

1. Gazzo A., Raimondi D., Daneels D., Moreau Y., Smits G., Van Dooren S., Lenaerts T. (2017) Understanding mutational effects in digenic diseases. *Nucleic Acids Research*. 45(15):e140.
2. Versbraegen N., Fouché A., Nachtegael C., Papadimitriou S., Gazzo A., Smits G., Lenaerts T. (2019) Using game theory and decision decomposition to effectively discern and characterise bi-locus diseases. *Artificial Intelligence in Medicine*. 99:101690.

Albeit they are not using expression data, their work allows one to examine whether a particular combination of variants in a pair of genes is a monogenic+modifier combination, which would imply that they could in principle produce similar results. Maybe these references are also worthwhile to add and discuss in light of your results.

3. The sensitivity of the method remains a concern as it reveals that it is not really that good at identifying the modifiers among the non-modifiers (independent of whether this is better than random or not). This weakness and its implications should be explicitly be made clear in the discussion as to not give the wrong impression to people wishing to use GENDULF in their work.

4. I have also the impression that the high specificity (success in identifying true negatives) is due to the fact that your data is unbalanced (many more cases that are not known to be not modifiers than modifiers), so classifying negative ones becomes very easy and thus specificity is high. Can you provide some information in your analysis on how many positive and negative instances are part of your sensitivity analysis? Can you correct for this unbalanced property?

5. I did not find the description on the randomization you used for the sensitivity analysis. I assume that you took a random sample of equal size as in step 2 from the results produced by step 1? Please clarify this in the methods section.

Reviewer #3:

The authors have done substantive revisions, and the manuscript and its clarity has improved quite a bit. I now understand the method and analyses much better. Most of my comments were addressed, but a few remain concerning the new analyses:

It's good that the sensitivity and specificity analysis was added, but the specificity metric in itself is not informative for such a biased data set (there's a huge number of genes that are not modifiers and not detected by GENDULF, hence specificity will be extremely high no matter what). Positive predictive value should be reported as well, since that is informative of how many genes detected by GENDULF are actual modifiers (it would be $8/132 = 6\%$ for lung CF). Obviously there are (hopefully!) new modifiers on the list, but since this data set is used for benchmarking, it should be used to its full extent, and the PPV is very important for downstream users to know.

Regarding the power analysis: the PPV framework would, in my opinion, be much more useful here than the current calculation. Since Step 2 is for exclusion of genes where the diseased individuals have lower expression than controls compared to the difference in the population sample, wouldn't the signature of a poorly powered Step 2 be exclusion of very few genes and low PPV, but not necessarily a much lower sensitivity? If I understand correctly, the current power calculation is looking at something different, and using a sensitivity-like test to see if a simulated DPM is discovered. I strongly suggest adding a PPV-type metric in this analysis. Also, I suggest adding more detail in the legend of Fig EV3 - it was difficult to understand.

In Fig EV6, I don't understand the scale of values between 0 and 1 for $-\log p$ -values. The base of the logarithm is not given, but if it's 10, that would mean that the p-values range between 1 and 0.1?? This is not consistent with any of the other data provided. P-value histograms are typically not in log scale and nominal values should be used. I'm confused and can't really review the content.

Minor:

In the second one of the following sentences, it should be colon, right? "We found that 8 of the 10 previously published CF lung modifiers that passed GENDULF step 1 also passed GENDULF step 2" ... "All four previously published lung CF modifiers that passed GENDULF step 1 were eligible at step 2 and also passed step 2."

Please add the unit of expression levels in Fig. 2-3

It's ExAC, not EXaC

Reviewer #1:

I thank the authors for providing answers to resolve my earlier concerns. Below some remaining remarks/concerns.

You mention an update of the literature review in your answer to my question concerning epistasis prediction. I'm therefore surprised you did not include the following references

1. Gazzo A., Raimondi D., Daneels D., Moreau Y., Smits G., Van Dooren S., Lenaerts T. (2017) Understanding mutational effects in digenic diseases. *Nucleic Acids Research*. 45(15):e140.
2. Versbraegen N., Fouché A., Nachtegaele C., Papadimitriou S., Gazzo A., Smits G., Lenaerts T. (2019) Using game theory and decision decomposition to effectively discern and characterise bi-locus diseases. *Artificial Intelligence in Medicine*. 99:101690.

Albeit they are not using expression data, their work allows one to examine whether a particular combination of variants in a pair of genes is a monogenic+modifier combination, which would imply that they could in principle produce similar results. Maybe these references are also worthwhile to add and discuss in light of your results.

Response: We thank reviewer 1 for pointing us to these interesting papers. We have cited both in the revised manuscript. These two papers are concerned with classifying published two-gene interactions into different scenarios, including primary gene + modifier, true digenic inheritance, and composite inheritance, rather than with finding previously unrecognized gene modifiers via a genome-wide search. As such, we find these methods and papers most relevant to our literature search of diseases to which GENDULF might be applied. If at least one established gene combination is classified by the methods of Gazzo et al. or Versbraegen et al. as what we called GCD (gene causing disease) + PM (potential modifier), then that characterization gives more hope that there may be additional modifiers (PMs) that can be

found using GENDULF for the same disease. This is now briefly discussed and mentioned accordingly on page 25 of the main text:

“The relevant literature includes instances of a GCD with a modifier, instances of true digenic inheritance where neither gene alone is sufficient to cause the full disease, and other more complicated two-gene scenarios. When there are sufficiently many examples, these three scenarios can be formally distinguished with machine learning techniques (Gazzo et al. 2017; Versbraegen et al. 2019).”

The sensitivity of the method remains a concern as it reveals that it is not really that good at identifying the modifiers among the non-modifiers (independent of whether this is better than random or not). This weakness and its implications should be explicitly be made clear in the discussion as to not give the wrong impression to people wishing to use GENDULF in their work.

Response: We agree with this concern – thank you. We now explicitly discuss the low sensitivity in the Discussion section, and make it clear for future users of GENDULF [page 24]:

“It is important to note that the sensitivity of GENDULF is generally low, even if substantially higher than random. Incorporating a disease-specific step into the framework of GENDULF, as done in the SMA analysis, increased the sensitivity of the predicted sets and yielded a manageable list of strong candidates. This should be done whenever possible. Regardless, GENDULF predictions should be followed by experiments to test the emerging candidates and validate predictions.”

I have also the impression that the high specificity (success in identifying true negatives) is due to the fact that your data is unbalanced (many more cases that are not known to be not modifiers than modifiers), so classifying negative ones becomes very easy and thus specificity is high. Can you provide some information in your analysis on how many positive and negative instances are part of your sensitivity analysis? Can you correct for this unbalanced property?

Response: We appreciate this comment. Indeed, the high specificity is principally due to the fact that only a small percentage of all genes are modifiers for CF or SMA (or likely any disease). The sensitivity is low partly because at least some modifiers have not yet been discovered. Focusing this discussion on CF, which is probably the best characterized disorder in terms of known modifiers, we apply the sensitivity and specificity analysis to all genes with GTEx expression for lung tissue, where the known modifiers (positives) considered are the 33 genes provided in dataset EV2. Hence, naturally, the specificity is very high (but notably higher than

would be expected by chance). The sensitivity is over 200-fold increase from the random rate. In practice, given 33 curated modifiers that have been originally found in the literature, the 10 and 8 modifiers predicted by GENDULF steps 1 and 2 (and additional two, *KRT8* and *MUC1* which were identified as modifiers by the GENDULF algorithm, and later found to be reported as modifiers in “Gene modifiers of cystic fibrosis lung disease: A systematic review”, *Pediatric Pulmonology* 2019; 54: 1356-1366, which we cite) provide a manageable size to be tested experimentally, and substantially improves the chances of identifying new modifiers. The unbalanced sizes of modifiers vs non modifiers is simply a characteristic of the data, which cannot be corrected without additional information that would allow focusing the initial search on a smaller set of genes, instead of starting with a genome-wide search. This is now further clarified in the revised main text [page 27]:

“The specificity is very high and close to the perfect 1.0, but this is partially driven by the small number of known modifiers (positives). The sensitivity is not very high, but still over two orders of magnitude higher than would be expected by chance, providing a manageable number of modifiers that are predicted with GENDULF. ”

I did not find the description on the randomization you used for the sensitivity analysis. I assume that you took a random sample of equal size as in step 2 from the results produced by step 1? Please clarify this in the methods section.

Response: We thank the reviewer for this comment. In performing the sensitivity analysis, we sampled groups that match the size of the case-control data in GENDULF step two. This is now clarified in the revised methods section [page 33]:

“To evaluate the sensitivity of GENDULF step 2, we sampled sets of patients with similar sizes as the case-control data, in the same manner described for the empirical P-value calculation for GENDULF step 2.”

Reviewer #3:

The authors have done substantive revisions, and the manuscript and its clarity has improved quite a bit. I now understand the method and analyses much better. Most of my comments were addressed, but a few remain concerning the new analyses:

It's good that the sensitivity and specificity analysis was added, but the specificity metric in

itself is not informative for such a biased data set (there's a huge number of genes that are not modifiers and not detected by GENDULF, hence specificity will be extremely high no matter what). Positive predictive value should be reported as well, since that is informative of how many genes detected by GENDULF are actual modifiers (it would be $8/132 = 6\%$ for lung CF). Obviously there are (hopefully!) new modifiers on the list, but since this data set is used for benchmarking, it should be used to its full extent, and the PPV is very important for downstream users to know.

Response: We appreciate this suggestion. Following up on it, we now added an analysis presenting the positive predictive value (PPV) of GENDULF to Appendix Figure S1 (was S2 in the previous version), and now refer to it from the main text (page 10 for Results and pages 33-34 for Methods). Indeed, there is more than a 2-fold increase of PPV from GENDULF step 1 to step 2, with only a small difference in the sensitivity and specificity values. The PPV with the 0.1 threshold used is 6.8%, as it also includes the gene *KRT8*, which was predicted by GENDULF, but not included in the original literature-based collection of modifiers. We expect that additional modifier genes within this list would be validated by other methods, especially wet lab experiments, in the future.

Regarding the power analysis: the PPV framework would, in my opinion, be much more useful here than the current calculation. Since Step 2 is for exclusion of genes where the diseased individuals have lower expression than controls compared to the difference in the population sample, wouldn't the signature of a poorly powered Step 2 be exclusion of very few genes and low PPV, but not necessarily a much lower sensitivity? If I understand correctly, the current power calculation is looking at something different, and using a sensitivity-like test to see if a simulated DPM is discovered. I strongly suggest adding a PPV-type metric in this analysis. Also, I suggest adding more detail in the legend of Fig EV3 - it was difficult to understand.

Response: We appreciate this comment. Indeed, the power analysis added in the first revision follows the standard definition and thus focuses on the probability of avoiding type 2 error. Hence, it evaluates a sample size that would allow high probability to correctly identify a true DPM, rejecting the null hypotheses when the alternative hypothesis is true.

We agree that maximizing the PPV, which corresponds to reducing or avoiding type 1 error, is a worthy objective. Thus, in planning GENDULF applications one may like to know what is the probability that the type 1 error is at most $p\%$ if sufficient samples are collected for the case-control analysis at step 2. Therefore, we acted on the reviewer's excellent suggestion by adding a "type 1 error analysis" to the GENDULF code, which is described in the revised Methods section. (Because the Editor asked us to make the Methods structured (see above), the context

on page 32 in which we made this important addition looks superficially different than in our first resubmission.) Reassuringly, we find that the type 1 error is below 5% for any number of case and control samples considered. We added the following text on pages 33-34 and modified the GENDULF software accordingly.

“In addition to the standard power analysis that estimates the number of cases and controls that would limit type 2 error, GENDULF can estimate the number of cases and controls that would limit type 1 error, and thereby improve its positive predictive value (PPV). To evaluate the “prospective PPV”, we consider a situation in which the null hypothesis should not be rejected; i.e., we want to avoid a false positive that would decrease the PPV. This estimation of prospective PPV differs fundamentally from the calculation of “retrospective PPV” in Appendix Figure S1C because the prospective calculation looks only at hypothetical expression data to determine what are true positive replicates and false positive replicates. In contrast, for the retrospective analysis in Appendix Figure S1C, the true positives are the predicted modifiers that have been reported in the literature and the false positives are predicted modifiers that have not been reported in the literature. The primary motivation for new methods such as GENDULF is the general awareness that many modifier genes still remain to be found, given our yet limited knowledge of the biology of many monogenic disorders. Hence, the retrospective PPV for the reported modifiers as shown in Appendix Figure S1 is expected to be low, even if the hypothetical PPV based on simulations may be high.

As for the power estimation, we sample N cases and N controls. For this type 1 error estimation, we take N samples of the GCD expression from S_{GL} and N samples from S_{GH} and evaluate whether the null hypothesis should be rejected with each sample. We do this sampling for all PMs of a given tissue and GCD (obtained from GENDULF step 1). Then, we compute what is the probability among all PMs that the type 1 error is below 0.05. We found that for any number of samples considered in Appendix Figure S2, the probability is greater than 0.95 that the type 1 error is below 0.05.”

The issue of how scientific predictions get validated and hence what we called “retrospective PPV” increases over time is an important question in history of science but, understandably, is far outside the scope of our genetics study.

In Fig EV6, I don't understand the scale of values between 0 and 1 for $-\log$ p-values. The base of

the logarithm is not given, but if it's 10, that would mean that the p-values range between 1 and 0.1?? This is not consistent with any of the other data provided. P-value histograms are typically not in log scale and nominal values should be used. I'm confused and can't really review the content.

Response: We apologize for the confusion and thank the referee for pointing this out. In an internal version of this figure, we used the natural logarithm scale of the P-values for this plot. However, the submitted version of the figure shows the nominal P-values, as the referee suggests. We corrected the error in the revised manuscript. Due to a request from the Editor (see above), the identifier of this figure is now S6 instead of the previous EV6.

Minor:

In the second one of the following sentences, it should be colon, right? "We found that 8 of the 10 previously published CF lung modifiers that passed GENDULF step 1 also passed GENDULF step 2" ... "All four previously published lung CF modifiers that passed GENDULF step 1 were eligible at step 2 and also passed step 2."

Response: We sincerely thank the referee for noticing this error. It has been corrected on page 13 in the revised manuscript.

Please add the unit of expression levels in Fig. 2-3

Response: Thank you. We added the unit of expression, for figures 2 and 3, which is RPKM for GTEx data, GPL96 microarrays for the CF case-control data, and FPKM for the SMA case-control data. To further clarify, we added to the caption of figure 3 the following sentence:

“The Reads Per Kilobase Million (RPKM) measure was used in the GTEx dataset, and the Fragments Per Kilobase Million (FPKM) measure was used in the SMA case-control dataset.”

It's ExAC, not EXaC

Response: Thank you. This has been corrected in the revised manuscript on page 20.

Thank you again for performing the last requested edits. We are now satisfied with the modifications made and I am pleased to inform you that your paper has been accepted for publication.

Corresponding Author Name: Eytan Ruppin

Manuscript Number: MSB-20-9701